# The landscape of genomic structural variation in Indigenous Australians

Andre L. M. Reis[1,2,3], Melissa Rapadas[1,2], Jillian M. Hammond[1,2], Hasindu Gamaarachchi[1,2,4], Igor Stevanovski[1,2], Meutia Ayuputeri Kumaheri[1,2], Sanjog R. Chintalaphani[1,2,3], Duminda S. B. Dissanayake[5,6], Owen M. Siggs[1,2,7], Alex W. Hewitt[8], Bastien Llamas[5,9,10,11], Alex Brown[5,11], Gareth Baynam[12,13,14], Graham J. Mann[5], Brendan J. McMorran[5], Simon Easteal[5], Azure Hermes[5], Misty R. Jenkins[15], The National Centre for Indigenous Genomics*, Hardip R. Patel[5✉] & Ira W. Deveson[1,2,3✉]

Indigenous Australians harbour rich and unique genomic diversity. However, Aboriginal and Torres Strait Islander ancestries are historically under-represented in genomics research and almost completely missing from reference datasets[1–3]. Addressing this representation gap is critical, both to advance our understanding of global human genomic diversity and as a prerequisite for ensuring equitable outcomes in genomic medicine. Here we apply population-scale whole-genome long-read sequencing[4] to profile genomic structural variation across four remote Indigenous communities. We uncover an abundance of large insertion–deletion variants (20–49 bp; $n = 136,797$), structural variants (50 b–50 kb; $n = 159,912$) and regions of variable copy number (>50 kb; $n = 156$). The majority of variants are composed of tandem repeat or interspersed mobile element sequences (up to 90%) and have not been previously annotated (up to 62%). A large fraction of structural variants appear to be exclusive to Indigenous Australians (12% lower-bound estimate) and most of these are found in only a single community, underscoring the need for broad and deep sampling to achieve a comprehensive catalogue of genomic structural variation across the Australian continent. Finally, we explore short tandem repeats throughout the genome to characterize allelic diversity at 50 known disease loci[5], uncover hundreds of novel repeat expansion sites within protein-coding genes, and identify unique patterns of diversity and constraint among short tandem repeat sequences. Our study sheds new light on the dimensions and dynamics of genomic structural variation within and beyond Australia.

Australia is home to hundreds of Aboriginal nations or clans who inhabited all geographical regions throughout the continent, prospering in their diverse environments for at least 50,000 years[6–10]. More than 250 distinct languages were spoken at the time of invasion by people from Europe and around 150 of these survive today[11]. Australian Indigenous communities practice cultures that are among the world's oldest continuous surviving cultures. These are highly varied, but commonly emphasize the importance of kinship, ancestry and relationships to the landscape and environment[10].

Whereas the remarkable cultural and linguistic diversity of Indigenous Australians is well documented, their rich and unique genomic diversity is relatively unexplored. Indigenous peoples have been historically under-represented in genomics research globally and Aboriginal ancestries are currently absent from leading international genomics resources, including the 1000 Genomes Project and gnomAD reference databases[1,2], as well as the recent draft Human Pangenome Reference[3]. Such resources are central to the interpretation, diagnosis and treatment of genetic disease, but have reduced utility for communities without appropriate representation[12]. There is a pressing need to close this Indigenous representation gap to ensure equitable outcomes from genomic medicine in Australia[13,14]. Moreover, as one of the six inhabited continents on earth, the current lack of genomic data from Australia is a major gap in our understanding of global human genomic variation.

[1]Genomics and Inherited Disease Program, Garvan Institute of Medical Research, Sydney, New South Wales, Australia. [2]Centre for Population Genomics, Garvan Institute of Medical Research and Murdoch Children's Research Institute, Darlinghurst, New South Wales, Australia. [3]Faculty of Medicine, University of New South Wales, Sydney, New South Wales, Australia. [4]School of Computer Science and Engineering, University of New South Wales, Sydney, New South Wales, Australia. [5]National Centre for Indigenous Genomics, John Curtin School of Medical Research, Australian National University, Canberra, Australian Capital Territory, Australia. [6]Institute for Applied Ecology, University of Canberra, Canberra, Australian Capital Territory, Australia. [7]Department of Ophthalmology, Flinders University, Bedford Park, South Australia, Australia. [8]Menzies Institute for Medical Research, University of Tasmania, Hobart, Tasmania, Australia. [9]Australian Centre for Ancient DNA, School of Biological Sciences and Environment Institute, University of Adelaide, Adelaide, South Australia, Australia. [10]ARC Centre of Excellence for Australian Biodiversity and Heritage, University of Adelaide, Adelaide, South Australia, Australia. [11]Indigenous Genomics, Telethon Kids Institute, Adelaide, South Australia, Australia. [12]Telethon Kids Institute and Division of Paediatrics, Faculty of Health and Medical Sciences, University of Western Australia, Perth, Western Australia, Australia. [13]Genetic Services of Western Australia, Western Australian Department of Health, Perth, Western Australia, Australia. [14]Western Australian Register of Developmental Anomalies, Western Australian Department of Health, Perth, Western Australia, Australia. [15]Immunology Division, The Walter and Eliza Hall Institute of Medical Research, Parkville, Victoria, Australia. *A list of authors and their affiliations appears at the end of the paper. ✉e-mail: hardip.patel@anu.edu.au; i.deveson@garvan.org.au

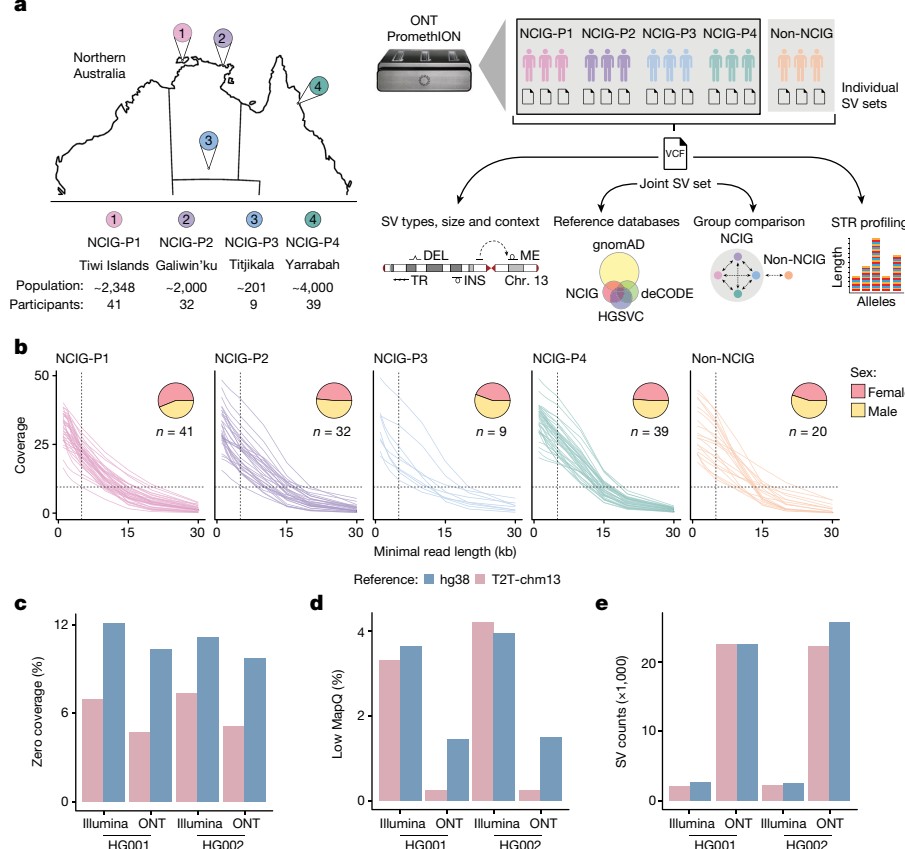

**Fig. 1 | Long-read sequencing in Indigenous Australian communities.**
**a**, Study design and analysis workflow. DNA samples were collected from four Indigenous communities: Tiwi Islands (NCIG-P1), Galiwin'ku (P2), Titjikala (P3) and Yarrabah (P4), and from unrelated European individuals (non-NCIG). The map shows geographic locations, with population sizes and participant numbers underneath. ONT sequencing was performed and reads aligned to the T2T-chm13 genome. SVs were called for each individual, then joint calling was performed to generate a non-redundant set of SVs, genotyped for each individual. SVs were characterized by type, size and context and compared to existing SV datasets. SVs were compared between individuals and communities, with non-NCIG individuals as an outgroup. Short tandem repeat (STR) alleles were genotyped to assess variation. Chr, chromosome; DEL, deletions; INS, insertions; ME, mobile elements. **b**, Average genomic coverage as sequencing reads were filtered by a minimum read-length cut-off. Each line represents one individual. Pie charts show the proportion of male and female participants from each community. **c**, Percentage of genome with zero coverage for Illumina short-read and ONT long-read libraries from HG001 and HG002, aligned to either hg38 or T2T-chm13. **d**, Percentage of genome covered by alignments with low mapping quality (MAPQ < 5). **e**, Number of SVs detected.

The National Centre for Indigenous Genomics (NCIG) aims to address this gap, by engaging Aboriginal and Torres Strait Islander communities in genomics research (https://ncig.anu.edu.au/). The NCIG has developed frameworks for Indigenous genomics that prioritize community leadership, participation and data sovereignty[15,16]. The NCIG aims to develop relationships of deep trust with Indigenous communities, cataloguing their genomic diversity and conducting research in a manner that is sustainable, ethical, and not only beneficial to the partner communities but also aligned with their own ways of knowing[15,16].

In this study, we performed population-scale long-read sequencing using Oxford Nanopore Technologies (ONT) instruments for four NCIG-partnered Aboriginal communities across northern and central Australia, as well as non-Indigenous Australians. The use of long-read sequencing technology, in combination with the recently completed telomere-to-telomere human reference genome[17] (T2T-chm13), enables us to explore uncharted Aboriginal genomic variation. Long reads can resolve repetitive or non-unique genes and regions that are intractable with dominant short-read sequencing platforms[18]. Long reads are also superior for the detection of structural variants (SVs), which account for the majority of the differences between the genomes of any 2 individuals and at least 25% of their deleterious alleles, yet are poorly understood owing to technical and analytical limitations[18–20].

Here we apply long-read sequencing[4] at population scale to a diverse Indigenous cohort. We begin to describe the landscape of genomic structural variation in Indigenous Australians and establish frameworks for interpreting this variation in the context of genomic medicine.

## ONT sequencing in Aboriginal communities

To explore genomic structural variation in Indigenous Australians, we performed whole-genome ONT sequencing on individuals from four remote Aboriginal communities with whom the NCIG has developed partnerships: Tiwi Islands (Wurrumiyanga, Pirlangimpi and Millikapiti communities; NCIG-P1), Galiwin'ku (NCIG-P2), Titjikala (NCIG-P3) and Yarrabah (NCIG-P4). These span a wide geographic, cultural and linguistic landscape (Fig. 1a). We sequenced between 9 and 41 individuals from each community, 121 in total. We also sequenced 18 non-Indigenous Australian individuals of European ancestry for comparison, and two reference individuals from the Genome in a Bottle project[21] (HG001, HG002; Supplementary Table 1). High molecular weight DNA was extracted from saliva or blood and sequenced on an ONT PromethION device (see Methods). Non-human reads present in saliva samples were identified and showed negligible rates of erroneous alignment to human chromosomes (Extended Data Fig. 1a,b). We obtained a median of 30-fold (range 14–47) genome coverage and 9.2 kb (2.7–16.8 kb)

read length N50 (Extended Data Fig. 1c,d). Although DNA samples varied in quality, we obtained a minimum of 10-fold coverage in reads of at least 5 kb for every individual, providing a strong foundation for profiling genomic structural variation across the cohort (Fig. 1b and Extended Data Fig. 1e).

Publication of the first complete human genome[17] was a landmark for the field, but so far there are few major studies outside the T2T Consortium that have used T2T-chm13 as their chosen reference genome. We evaluated mappability and structural variant (SV) detection against the T2T-chm13 reference, by comparison to hg38, using both ONT and short-read sequencing data from the HG001 and HG002 reference samples (Methods). As expected, ONT data exhibited superior unique-alignment coverage and more comprehensive SV detection than short reads (Fig. 1c–e). These advantages were further enhanced by use of the T2T-chm13 reference, which had proportionally fewer regions of zero coverage (mean 4.9% versus 10.0%) or low mappability (MAPQ < 5; mean 0.2% versus 1.4%) and, as a result, afforded an additional approximately 125 Mb of total reference sequence that was accessible to analysis with ONT data (Fig. 1c,d). ONT sequencing depth had negligible effect on SV detection above a threshold of approximately 20X coverage, consistent with independent benchmarking[22] (Extended Data Fig. 2a). Manual inspection of medically relevant repetitive genes, such as *MUC1*[23] (Extended Data Fig. 2b), showed these were generally best resolved using the combination of ONT and T2T-chm13. Together, these results highlight the advantages of long-read sequencing and T2T-chm13 for profiling genomic structural variation at high resolution.

Adopting T2T-chm13 as our genome reference, we called large insertion–deletions (indels) (20–49 bp) and SVs (50 bp–50 kb) in each individual (CuteSV[24]; Fig. 1a). Variants were filtered to exclude events with weak evidence (QUAL ≥ 5). We detected 21,723 SVs, on average, per individual, of which 19,089 were retained after filtering (Extended Data Fig. 1f). The retained SV count is somewhat lower than reported in several recent long-read sequencing studies[25,26], reflecting our preference for retaining only high-confidence SVs. Callsets were then merged (using Jasmine[27]) into a unified joint-call catalogue comprising 159,912 unique SVs and 136,797 large indels (Fig. 1a and Methods). Notably, this surpasses the 134,886 SVs recently identified by ONT sequencing of 3,622 Icelanders[25] (the largest long-read sequencing study published to date), reflecting higher genetic heterogeneity in our smaller cohort. We additionally applied a read-depth method (CNVPytor[28]) to identify large copy number variants (CNVs) (>50 kb) in each individual. We identified 11 high-confidence ($P < 10^{-4}$) CNVs per individual (9 deletions, 2 duplications) on average, which were merged into 156 unique regions of variable copy number across the cohort (Extended Data Fig. 3a).

## Genomic structural variation landscape

To better characterize the landscape of genomic structural variation, we next stratified variants by type, size and context (Fig. 1a and Methods). A clear majority of all non-redundant variants (84.9%) were composed of repetitive sequences, including 103,425 STR (2–12 bp) and 123,667 tandem repeat (TR) (>12 bp) expansions and contractions, and 25,096 insertions or deletions of interspersed mobile element sequences (Fig. 2a). The remaining 37,574 variants (12.6%) were found to be non-repetitive (Fig. 2a). Although CNVs were few in number, they encompassed >65 Mb of genome sequence across the entire cohort, with an average of 2.8 Mb per individual (Extended Data Fig. 3b). We observed deletions up to 13 Mb (mean 243 kb) and duplications up to 1.8 Mb (mean 303 kb; Extended Data Fig. 3c,d).

Structural variation was not distributed evenly across the genome, but showed higher density within approximately 5 Mb of the telomere on each chromosome (Fig. 2b), as has been reported elsewhere[25,29]. This effect was almost entirely driven by TR-associated SVs, which were enriched in sub-telomeric regions, with other classes being evenly

distributed (Extended Data Fig. 4a). Both metacentric and acrocentric chromosomes were similarly affected (Extended Data Fig. 4b).

We observed characteristic differences in size between variants of different types (Fig. 2c,d). TR-associated SVs were generally larger than STRs or non-repetitive SVs. Size distributions for mobile element SVs displayed clear peaks around expected sizes for major repeat families, including Alu (a SINE of approximately 280 bp), L1 (a LINE of approximately 6 kb) and SVA[30] (SINE-R/VNTR/Alu; a retroposon of approximately 2 kb) (Fig. 2c,d). Whereas most SVs associated with mobile elements encompassed only part of an annotated element, the aforementioned peaks are formed by SVs encompassing one or more complete elements, which represent transposition events occurring since the common ancestor of individuals in our cohort and the T2T-chm13 reference (European origin; Fig. 2d). Among these complete elements, SINEs were dominant ($n = 8,867$), reflecting comparatively high Alu activity, and significant numbers of LINE ($n = 501$) and retroposon ($n = 327$) transpositions were also detected (Fig. 2a,d).

Given the inclusion of unique, under-represented Australian communities and the use of long-read sequencing, our catalogue contained a high proportion of SVs that have not been previously annotated (Fig. 2e). We compared our SV callset to: (1) the gnomAD SV database[19], which spans a diverse global cohort sequenced on short-read platforms; (2) an SV annotation published recently by deCODE genetics[25], based on population-scale ONT sequencing of Icelandic individuals; and (3) a state-of-the-art long-read SV annotation based on 35 diverse individuals analysed by the Human Genome Structural Variation Consortium (HGSVC). For this analysis, it was necessary to first convert SV coordinates to the hg38 reference, on which these annotations are based (using LiftOver). A significant number of SVs could not be lifted from T2T-chm13 to hg38 because their corresponding positions were fully (24.9%) or partially (18.3%) missing from the latter (Fig. 2e). Of the 90,578 out of 159,912 SVs that were successfully lifted to hg38, we found a highly similar annotated SV (more than 80% reciprocal overlap) for 37,421 SVs and an annotated SV at the same position with moderate similarity (50–80%) for 22,625 SVs (Fig. 2e). The latter were especially common for TR- and STR-associated SVs, where alternative alleles at variable TR or STR loci often appear as partially overlapping SVs (Extended Data Fig. 4c,d). Together, this shows that there is an annotated SV for around 38% of all non-redundant SVs in our callset (Fig. 2e), with the remaining SVs that were successfully lifted having low (<50%; $n = 8,770$) or no (0%; $n = 21,762$) overlap with any annotated variant. Because SVs that could not be lifted to hg38 cannot be assessed in this manner, we instead provide an upper-bound novelty estimate of 62% (assuming non-lifted SVs are all novel) and a lower-bound estimate of 19%.

## Distribution and diversity

We next assessed the distribution of genomic structural variation among Indigenous and non-Indigenous individuals in the cohort. Overall, the majority of all non-redundant SVs were either private (that is, found in a single individual; 26.3%) or polymorphic (found in less than 50% of individuals; 65.5%), with the remaining being classified as major (found in more than 50% of individuals; 7.8%) or shared alleles (found in all individuals; 0.2%) (Extended Data Fig. 5b). Although different SV types were distributed uniformly among individuals and communities (Fig. 3a and Extended Data Fig. 5a), they varied in the degree to which they were shared between individuals (Fig. 3b and Extended Data Fig. 5c). For example, the proportional representation of STR- and TR-associated SVs was skewed towards polymorphic and private variation, whereas mobile elements and non-repetitive SVs were proportionally enriched among major and shared variation (Fig. 3b). These trends indicate the different rates at which different classes of SVs emerge and change over generations.

A large proportion of SVs across the complete non-redundant catalogue was seen only among Indigenous individuals (NCIG-only;

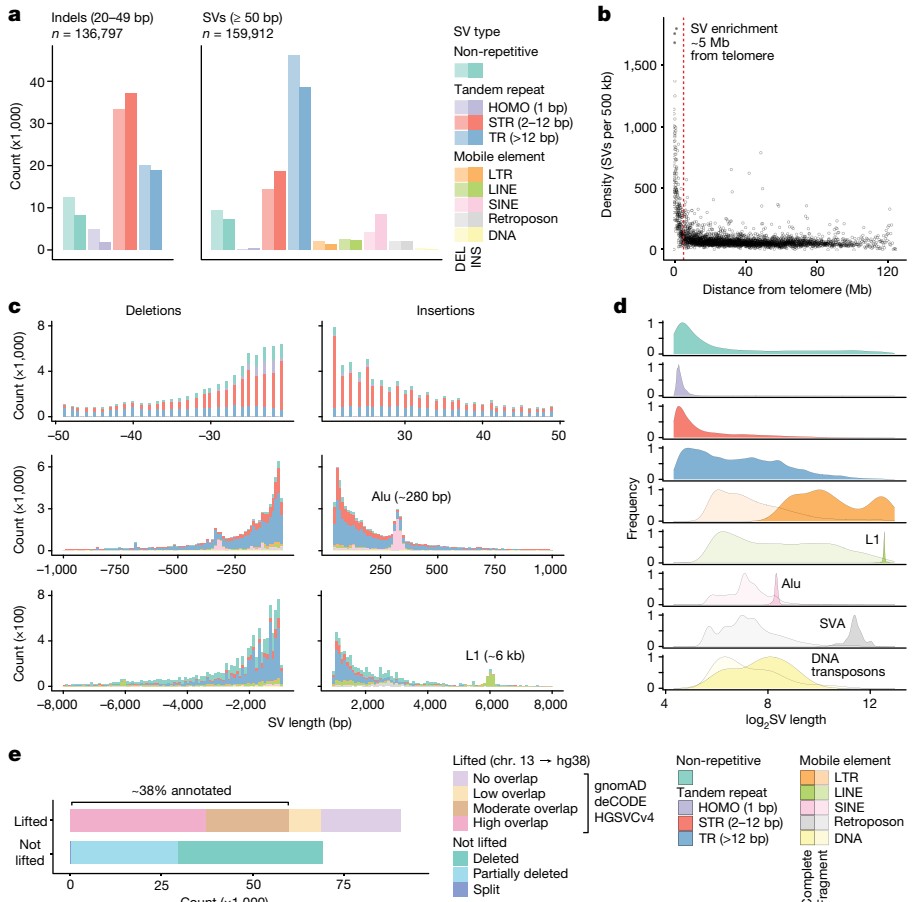

**Fig. 2 | Landscape of genomic structural variation. a**, Number of non-redundant variants identified across the cohort (*n* = 141). Large indels (left; 20–49 bp) and SVs (right; ≥50 bp) are shown separately and parsed by type: non-repetitive, tandem repeats (homopolymer (HOMO), STR and TR) and mobile elements (long terminal repeat (LTR), long interspersed nuclear elements (LINE), short interspersed nuclear element (SINE), retroposon and DNA/DNA transposon (DNA)). Light shades represent deletions and dark shades represent insertions. **b**, Frequency of non-redundant SVs relative to distance from the nearest telomere. **c**, Size distribution of insertions (positive values) and deletions (negative values), parsed by type (colour scheme as in **a**). Characteristic peaks for Alu elements (280 bp) and L1 elements (6 kb) are

marked. **d**, Size distributions for each variant type. Mobile element SVs are classified as 'complete' if they encompass one or more complete annotated element or 'fragment' if they encompass only part of an annotated element. **e**, Number of non-redundant SVs found in a search performed against a combination of the gnomAD, deCODE and HGSVC (freeze 4) SV annotations. SVs were first lifted from T2T-chm13 to hg38. Some could not be lifted because they were deleted or partially deleted in hg38. SVs that could be lifted were categorized as 'annotated' or 'unannotated' on the basis of reciprocal overlap to any single annotated SV. High (>80%) and moderate (50–80%) overlaps were considered as annotated, whereas low (<50%; beige) and no overlap were considered as unannotated.

48.5%) or only among non-Indigenous participants (NCIG-absent; 9.2%) (Fig. 3c). NCIG-only SVs made up a significantly higher proportion of total SVs in a given Indigenous individual (15.0 ± 2.0%) than NCIG-absent SVs in non-Indigenous individuals (5.2 ± 0.4%) (Fig. 3c and Extended Data Fig. 6a). The majority of NCIG-only variants were polymorphic (Fig. 3d and Extended Data Fig. 6b) and were previously unannotated—more so than for NCIG-absent variants (Extended Data Fig. 6c). On average, each Indigenous individual harboured 2,884 ± 520 NCIG-only SVs, of which 311 ± 259 were unannotated (lower-bound estimate) and may therefore represent exclusively Australian Indigenous variation.

The clear genetic distinctions between Indigenous Australian and non-Indigenous Australian individuals was further reiterated by principal coordinate analysis (PCOA) and fixation index ($F_{ST}$) analysis of structural variation (Fig. 4a and Extended Data Fig. 7a). This also highlighted the distinct genetic architecture of different communities, which formed largely separate PCOA clusters (Fig. 4a). Indeed, among Indigenous individuals, we found that 56.4% of NCIG-only SVs were found in just a single individual or community, whereas NCIG-only SVs shared between more than one individual across all communities were relatively rare (2.8%) (Fig. 4b and Extended Data Fig. 7b). This was

corroborated by independent analysis of SNVs detected with short-read sequencing data from the same NCIG partner communities, with similar proportions of continent- and community-specific variation being observed (Extended Data Fig. 7c,d). Shared SVs showed a proportional enrichment of mobile elements and a depletion of TR SVs (Extended Data Fig. 8a), consistent with the contrasting polymorphism for these SV types (Fig. 3b). Of the approximately 311 exclusively Indigenous SVs in a given individual identified above, an average of around 185 ± 31 were not found outside their community.

Next, we generated discovery curves that model the diversity of structural variation within a set of individuals (Methods). Across the 121 Indigenous individuals in the cohort, cumulative SV discovery did not approach saturation, indicating many further SV alleles remain to be sampled (Fig. 4d and Extended Data Fig. 8b–d). This diversity was not shared equally among different communities or variant types (Fig. 4c and Extended Data Fig. 8e). Individually, the Tiwi Islands (NCIG-P1), Galiwin'ku (NCIG-P2) and Titjikala (NCIG-P3) communities each showed lower within-community SV diversity than seen among the non-NCIG comparison group, reflecting their small population sizes and relative isolation (Fig. 1a). By contrast, Yarrabah (NCIG-P4) harboured

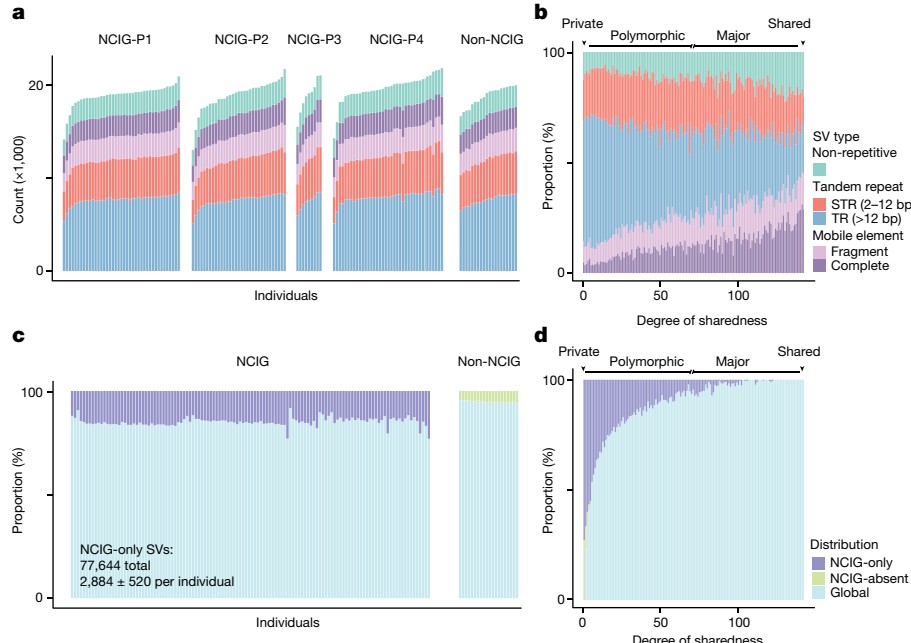

**Fig. 3 | Distribution of SVs in Indigenous and non-Indigenous individuals.**
**a**, Number of SVs identified in individuals from each group, parsed by type.
Colour scheme as in **b**. **b**, Proportional representation of different types for
SVs identified within a given number of individuals (degree of sharedness). SVs
were labelled as private (1 individual), polymorphic (more than one individual
and less than 50% of individuals), major (≥50%, but not all individuals) and

shared (all individuals). **c**, Proportion of SVs in each individual that were
found exclusively in NCIG individuals (NCIG-only), or exclusively in non-NCIG
individuals (NCIG-absent), or across both (global). Colour scheme as in **d**.
**d**, Proportional representation of NCIG-only, NCIG-absent and global SVs
according to degree of sharedness.

substantially higher genomic diversity than the other communities,
a higher proportion of private variation, and alone showed greater
diversity than the non-Indigenous group (Fig. 4c and Extended Data
Fig. 8b). NCIG-only SVs showed greater heterogeneity among NCIG
individuals than for NCIG-absent SVs among non-NCIG individuals
(Extended Data Fig. 8b). Finally, we found that SV diversity was driven
most strongly by TR-associated SVs, whereas new discovery of mobile
elements and non-repetitive SVs was largely saturated (Extended Data
Fig. 8b,d).

## Functional context

Given the vast diversity of genomic structural variation described above
and the predominance of SV classes that are poorly studied, we next
used measures of purifying selection to investigate their functional
relevance. A large indel or SV intersecting with one or more coding
sequence (CDS) exons in a protein-coding gene is likely to truncate or
alter its open reading frame, whereas a variant within an intron, untrans-
lated region (UTR) or proximal gene-regulatory region may affect tran-
scription, translation or splicing. The extent to which these events
disrupt gene function should be modelled by depletion of structural
variation within essential genes among otherwise healthy populations[19].

Across our complete cohort ($n$ = 141), we detected 126,473 non-
redundant variants (58,079 indels and 68,394 SVs) intersecting protein-
coding loci, including 1,462 affecting CDS exons. An average individual
possessed 156 ± 15 CDS variants and 20,124 ± 1,308 within non-CDS
regions of protein-coding genes (introns, UTRs and proximal regulatory
regions) (Extended Data Fig. 9a). There was an enrichment of private
variants intersecting CDS regions (33.7%) compared with the propor-
tion of private variants (24.1%) in intergenic regions, consistent with
purifying selection. Variants intersecting CDS exons were almost all
either non-repetitive (33.7%) or TR-associated (59.5%), with a strong
depletion of STR and mobile element SVs in CDS, relative to intronic
and intergenic regions (Fig. 5a). We also identified 82 large CNVs that

constituted whole-gene deletions ($n$ = 225 genes) or whole-gene dupli-
cations ($n$ = 237 genes) across the cohort (Extended Data Fig. 9b,c).

We next parsed protein-coding genes according to loss of function
observed/expected upper-bound fraction (LOEUF), a metric that
quantifies their intolerance to loss-of-function variation, developed
previously by gnomAD[2]. This approach revealed clear constraint on
structural variation in CDS regions (Fig. 5b). For example, we saw an
approximately 12-fold reduction in the size-normalized density of
structural variation among the most essential genes (LOEUF decile
1; $1.29 \times 10^{-5}$) compared with the least essential genes (LOEUF decile
10; $1.60 \times 10^{-4}$) (Fig. 5b and Methods). We also observed significant,
albeit weaker, constraint on non-CDS variants, with an approximately
1.5-fold difference in variant frequency between the highest and lowest
deciles (Fig. 5b). Mobile element-associated SVs showed the strongest
purifying selection (1.8-fold difference between highest and lowest
deciles) and STR-associated SVs showed the weakest (1.3-fold) (Fig. 5b).
Assessing LOEUF among Indigenous-specific variation, we found that
most SVs (81.1%) in CDS regions of essential genes were private or
community-specific (Extended Data Fig. 9d).

Since evidence of selection is critical for interpreting the functional
relevance of genetic variation, the findings above help to establish
the suitability of our analysis framework and SV catalogue to inform
genomic medicine applications in Indigenous Australians. Even among
just 141 individuals sequenced here, we identified 69 deletions or
insertions affecting CDS regions of genes in LOEUF deciles 1 and 2, 44
of which were not previously annotated. This includes complete or
near-complete deletions of essential genes including *PRKRA*, *BRD9*,
*SHOX*, *NID2*, *PABPC5*, *JAZF1*, *NIPA2* and *ANOS1*, and a large somatic
CNV deletion (13 Mb) in a non-NCIG individual affecting three genes
in LOEUF decile 1 (*ARHGAP42*, *YAP1* and *DCUN1D5*).

One notable SV in an essential gene (LOEUF decile 1) was a 130 bp
CAG STR expansion in *ATXN3* that is known to cause Machado–Joseph
Disease[31] (MJD, also known as spinocerebellar ataxia type 3). This
pathogenic allele was detected in a single individual from Galiwin'ku

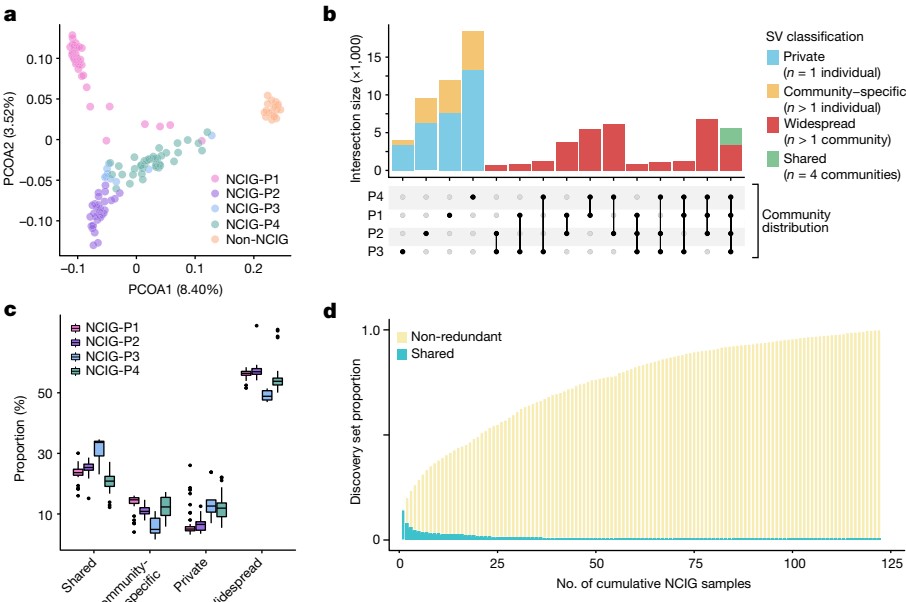

**Fig. 4 | Distribution of SVs between Indigenous communities. a**, PCOA representing the distance between individuals in the cohort based on their SV compositions. The percentage of variance is indicated in parentheses for each principal coordinate axis (PCOA1 and PCOA2). Individuals are coloured according to their group. **b**, Distribution of NCIG-only variants (Fig. 3) shared among the four NCIG communities. SVs were classified as private (*n* = 1 individual), community-specific (*n* > 1 individual in 1 community), widespread (*n* > 1 individual in more than 1 community) or shared (*n* > 1 individual in all 4 communities). **c**, Proportion of private, community-specific, widespread and shared NCIG-only variants among individuals, grouped by community. A total of *n* = 141 individuals (41 NCIG-P1, 32 NCIG-P2, 9 NCIG-P3, 39 NCIG-P4 and 20 non-NCIG individuals) were examined. The centre line shows the median, box edges delineate the interquartile range (IQR) and whiskers extend to 1.5× IQR from the hinge. **d**, SV discovery curve in which, starting with a single NCIG individual, the number of new non-redundant SVs is counted as new individuals are iteratively added. SVs shared among all previously added samples are shown as green portions of each bar.

community (NCIG-P2), with long-read sequencing clearly defining the size, position and sequence of the STR expansion, unlike short-read whole-genome sequencing on the same individual (Fig. 5c and Extended Data Fig. 9e). MJD is a late onset, progressive movement disorder with autosomal dominant inheritance and complete penetrance for expansions of this size[31]. MJD affects around 5 out of every 100,000 people worldwide, but is estimated to be more than 100 times more prevalent among Indigenous populations in some areas of Australia's Northern Territory[32].

The individual in question had consented to receive reportable genetic findings arising from NCIG research. This prompted an ongoing dialogue between NCIG, Galiwin'ku representatives, local genetic counsellors and the MJD foundation (https://mjd.org.au/), who work with remote Northern Territory Aboriginal communities to develop unique clinical genetics service models tailored for their needs[33]. Under recommendation of the MJD Foundation, genetic counsellors were able to contact the individual and their family, arranging for clinical testing and appropriate follow-up.

## STR expansions

The preceding example highlights the utility of long-read sequencing for profiling variation in STR sequences—both normal and pathogenic—as well as the importance of ancestry in interpreting this variation. STRs are highly polymorphic, and STR expansions are causative pathogenic variants in at least 37 neurogenic and 10 congenital disorders[5]. However, STR expansions are refractory to analysis with short-read sequencing and, as a result, have been relatively poorly characterized to date, particularly among minority communities, such as Indigenous Australians. Our dataset provides a unique opportunity to explore allelic diversity in STR sequences at high-resolution and population scale.

Across the cohort, we detected 55,595 non-redundant variants that constitute STR expansions (that is, insertions) and 47,830 STR contractions (that is, deletions; Methods) relative to the T2T-chm13 reference. These ranged in size from approximately 20 bp (our lower cut-off) to 99,204 bp in size and occurred predominantly in intergenic (63.9%) or non-CDS gene regions (35.9%) (Fig. 6a). STR period size was negatively correlated with the global frequency of STR expansions and contractions. However, trinucleotide and hexanucleotide repeats were outliers from this trend, showing markedly lower frequencies than other periods (Fig. 6a). The opposite was true within CDS regions, where in-frame expansions (that is, 3 bp, 6 bp, and so on) occurred at higher frequencies than other periods (Fig. 6a). Therefore, although in-frame expansions and contractions are more tolerated within coding sequences (because they do not cause frameshifts), there appears to be higher constraint on in-frame STRs across the remainder of the genome. We speculate that this acts to limit the potential for spurious expression of toxic homomeric polypeptides that contribute to pathogenicity in many STR disorders[5].

Consistent with this idea, expansions of the STR motifs associated with poly-glutamine (CAG) and poly-glycine (CGG) disorders, which include MJD and a range of other disorders, were globally rare by comparison to other motifs[5] (Fig. 6b). In contrast to these dominant gain-of-function disorders, the GAA expansion motif that triggers epigenetic silencing of *FXN* (that is, loss of function) in Fredrich's ataxia, was relatively common, suggesting expansions of this motif are not typically deleterious in other contexts[5] (Fig. 6b).

Besides triplets, intronic pentanucleotide STRs are most widely associated with known disorders[5]. In this context, it was notable that pentanucleotide STR expansions occurred at lower frequency (approximately 17%) within intronic regions than for other period sizes (around 40%), indicative of context-specific constraint (Fig. 6a). However, in contrast

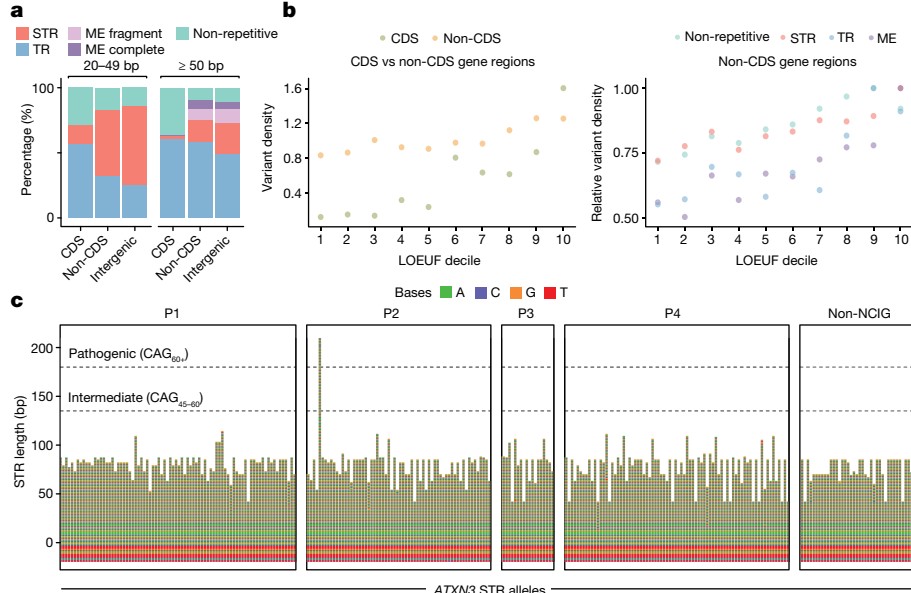

**Fig. 5 | Functional relevance of genomic structural variation. a**, Proportion of variant types identified within CDS exons, non-CDS regions of protein-coding genes (introns, UTRs and ±2 kb proximal regulatory regions) and intergenic regions, for large indels (left; 20–49 bp) and SVs (right; ≥50 bp). Variants are classified as: non-repetitive, STR, TR and mobile element (fragment or complete). **b**, Left, variant density in CDS and non-CDS regions of protein-coding genes in different LOEUF deciles, which quantify intolerance to loss-of-function variation (first decile corresponds to the highest constraint). **b**, Right, variant density in non-CDS parsed by variant type and normalized to their 10th decile. **c**, Sequence bar chart shows *ATXN3* STR alleles, including 20 bp of upstream flanking sequence, genotyped for every individual (two alleles each). Dashed lines show STR size cut-offs for intermediate and fully pathogenic alleles. A single pathogenic allele was identified in an NCIG-P2 individual. All others are in the 'normal' range.

to triplet disorders, motifs known to be associated with disease, such as AAGGG (CANVAS (cerebellar ataxia with neuropathy and vestibular areflexia syndrome)), TTTCA (FAME (familial adult myoclonic epilepsy)) and TGGAA (SCA31 (spinocerebellar ataxia type 31)), were among the most frequent pentanucleotide expansions (Fig. 6b).

To better resolve the STR landscape, we performed diploid STR genotyping, expansion discovery and visualization[34]. We focused our analysis on 685 STR sites of period 3 bp or larger within protein-coding loci (including 7 within CDS exons) that were significantly expanded in at least one individual (Extended Data Fig. 10a), as well as 50 known disease-associated STR loci[5]. Using this approach, we stratified STR sites according to allelic diversity across communities, identifying 231 sites (18.9%) that showed inter-community differences in allelic composition (Fig. 6c and Extended Data Fig. 10b). We found 155 sites that were more diverse in Indigenous than non-Indigenous individuals and 76 sites where the opposite was true (Fig. 6c). Many STRs showed more local effects, such as increased allelic diversity in just a single community, or expansions that were limited to a single individual or a small number of individuals (Fig. 6c).

## Discussion

Our study is a major genomic survey of Indigenous Australians. Previous publications that include genomic data from Aboriginal and/or Torres Strait Islander peoples have largely focused on historical demographic processes[8,35–38]. These relied on mitochondrial[38,39] or short-read whole-genome sequencing[8,35–37]. By contrast, we used whole-genome long-read sequencing to generate data that are suitable for exploring the landscape of Indigenous genomic structural variation.

We found a diversity of structural variation across four remote Aboriginal communities in northern and central Australia. This was predominantly repetitive, encompassing thousands of tandem repeats and mobile elements. A significant proportion was found only in Indigenous individuals in our study, and have not been previously annotated in diverse global reference data[19,25]. However, exclusively Indigenous variants were generally not shared throughout the continent.

Our study sheds new light on the rich and unique genetic diversity of Indigenous Australians. Owing to the long history of continuous occupation, Australia's Indigenous peoples are highly genetically distinct from non-Australians. This underscores the need for ancestry-appropriate reference data for genomic medicine, of which there is a shortage[12–14]. Moreover, Indigenous Australians should not be viewed as genetically homogenous. We show that different communities, clans and/or nations have highly distinct genomic architectures, mirroring their cultural and linguistic diversity[10]. Therefore, broad engagement—far beyond the four communities we have profiled here—will be required to adequately survey Indigenous genomic variation and, ultimately, to achieve equitable outcomes in genomic medicine.

Our study is among a small number of recent efforts to implement long-read sequencing at population scale[4], and others have so far focused on comparatively homogeneous, well-studied populations (for example, Icelandic[25], Chinese[40] and Japanese[41] populations). Although our cohort was smaller than each of these studies, we identified a greater number of total non-redundant SVs, reflecting higher genetic heterogeneity among our participants. We use this rich catalogue to articulate a number of fundamental insights into the landscape of genomic structural variation in human populations that reach beyond Australia. For example, we show that: (1) SVs are predominantly repetitive, with TRs, STRs and mobile elements underpinning around 87% of SVs per individual; (2) SVs of different types and sizes show clear differences in their dynamics of inheritance, with TR and STR SVs being more polymorphic than mobile elements and non-repetitive SVs; (3) SVs in both CDS and non-CDS regions of protein-coding genes are under purifying selection and different SV types show different signatures of constraint.

Our high-resolution survey of allelic diversity among STRs is similarly informative. We uncover an abundance of STR variation across the genome. STRs show distinct, context-specific signatures

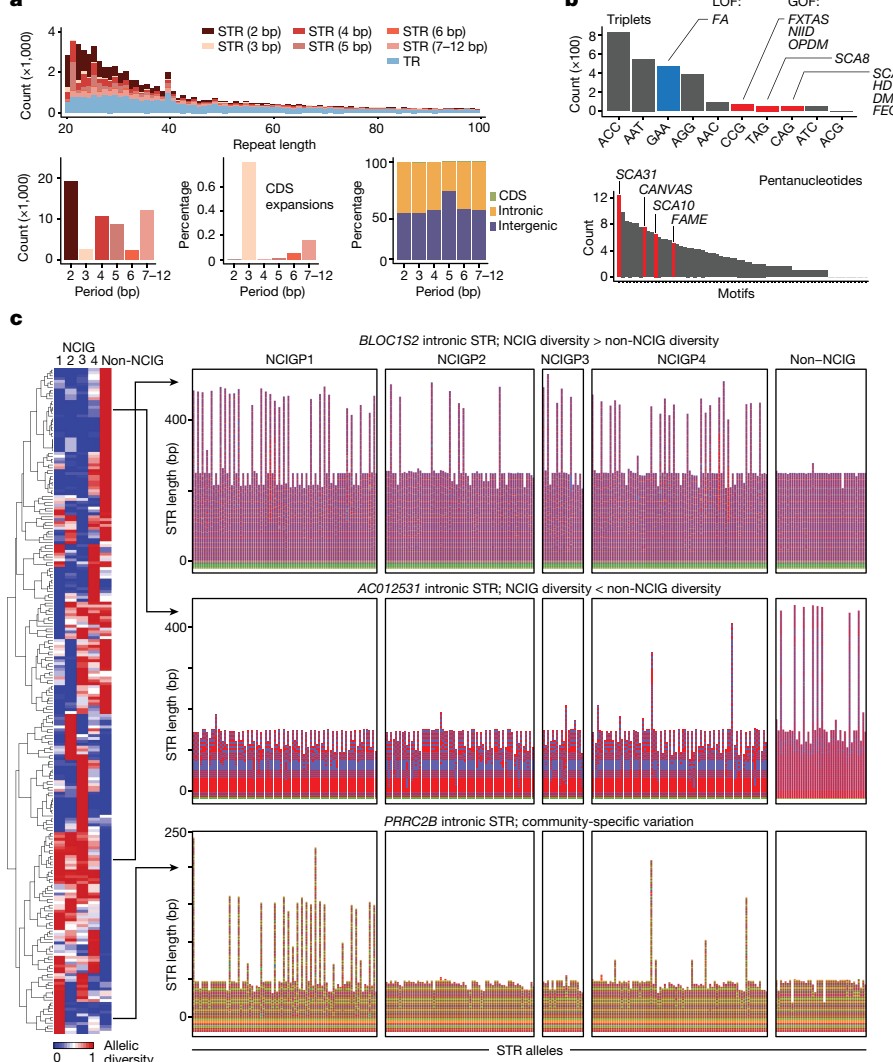

**Fig. 6 | Landscape of STR expansions. a**, Top, length distribution of all non-redundant TR and STR insertions (that is, expansions) detected across the cohort (*n* = 141), broken down by period size. Bottom left, number of non-redundant STR expansions in each period. Bottom centre, relative frequency of each period within CDS exons. Bottom right, proportional composition of each period based on genomic context. **b**, Frequency of every possible triplet (top) and pentanucleotide (bottom) motif among non-redundant STR expansions. Selected motifs known to cause different repeat disorders are highlighted. For triplet disorders, pathogenic motifs with gain-of-function mechanisms are shown in red, those with loss-of-function mechanisms are shown in blue. **c**, Left, normalized standard deviation (range 0–1) of allele sizes observed within each community group (matrix columns) for different STR sites (matrix rows). All expanded sites of period ≥3 bp within protein-coding genes, in which allelic composition was significantly different between groups are shown (one-way ANOVA, *P* < 0.05). Hierarchical clustering groups STR sites on the basis of patterns of variability between groups. **c**, Right, all STR alleles (two per individual) for three example sites showing distinct patterns of variation. *BLOC1S2* (top) has an intronic STR with higher allelic diversity in NCIG versus non-NCIG individuals. *AC012531* (middle) has lower allelic diversity in NCIG versus non-NCIG individuals. *PRRC2B* (bottom) shows community-specific patterns, with NCIG-P1 and NCIG-P4 exhibiting heterogeneity.

of selection, with specific periods and motifs showing elevated constraint, globally. For both novel expansion sites and known disease-associated loci, we show pervasive Indigenous versus non-Indigenous and inter-community differences in STR allele composition. Constructing a clear picture of this complex background of normal STR variation is critical for the discovery and diagnosis of STR expansion disorders[34]. However, most existing reference data are based on European and East Asian cohorts and are therefore of reduced suitability for Indigenous Australian communities. Tangible examples of local effects are provided by a unique STR motif that causes CANVAS in individuals of Māori descent[42] and MJD, an expansion disorder with markedly high frequency in Northern Territory Aboriginal communities[32]. Our study begins to establish the appropriate context for interpreting Indigenous STR variation in future genomic medicine initiatives. Indeed, the unexpected identification of a pathogenic MJD expansion in one individual highlights the strength of our approach in this domain.

## Ethics and inclusion

We are indebted to the individuals and their communities who participated in this research and to the NCIG Indigenous-majority Governance Board who helped guide this work in a culturally appropriate manner. The research was conducted in accordance with core principles of Indigenous community engagement, leadership and data sovereignty, as set out in the NCIG governance framework, approved under the Australian federal legislation (https://ncig.anu.edu.au/files/NCIG-Governance-Framework.pdf).

Saliva and/or blood samples were collected from consenting individuals among four NCIG-partnered communities: Tiwi Islands (comprising the Wurrumiyanga, Pirlangimpi and Millikapiti communities), Galiwin'ku, Titjikala and Yarrabah, between 2015 and 2019. This study was approved by the Australian National University Human Research Ethics Committee (Ethics protocol number 2015/065). Non-Indigenous comparison data, generated from unrelated Australian individuals of European ancestry, was drawn from two existing biomedical research cohorts: the Tasmanian Ophthalmic BioBank (Ethics protocol number 2020/ETH02479) and the Australian and New Zealand Registry of Advanced Glaucoma (Southern Adelaide Clinical Human Research Ethics Committee approval 305-08).

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

**The National Centre for Indigenous Genomics**

Glen Pearson[12], Yvette Roe[16,17], Janine Mohamed[18], Ben Murray[19], Lyndon Ormond-Parker[20,21], Gareth Baynam[12,13,14], Erica Kneipp[22], Keith Nugent[23] & Graham Mann[5]

[16]Molly Wardaguga Research Centre, Faculty of Health, Charles Darwin University, Brisbane City, Queensland, Australia. [17]Mater Research Institute and School of Nursing and Midwifery, University of Queensland, South Brisbane, Queensland, Australia. [18]The Lowitja Institute, Melbourne, Victoria, Australia. [19]Office of the Registrar of Indigenous Corporations, Canberra, Australian Capital Territory, Australia. [20]School of Media and Communication, RMIT University, Melbourne, Victoria, Australia. [21]Centre of Excellence for Automated Decision-Making and Society, RMIT University, Melbourne, Victoria, Australia. [22]Commonwealth Scientific and Industrial Research Organisation, Canberra, Australian Capital Territory, Australia. [23]The Australian National University, Canberra, Australian Capital Territory, Australia.

## Methods

### Cohorts

Saliva and/or blood samples were collected from consenting individuals among four NCIG-partnered communities: Tiwi Islands (comprising the Wurrumiyanga, Pirlangimpi and Millikapiti communities), Galiwin'ku, Titjikala and Yarrabah, between 2015 and 2019. Non-Indigenous comparison data, generated from unrelated Australian individuals of European ancestry, was drawn from two existing biomedical research cohorts: the Tasmanian Ophthalmic BioBank, and the Australian and New Zealand Registry of Advanced Glaucoma.

### Saliva sample collection and DNA extraction

Saliva samples were collected in Oragene DNA collection tubes (OG-500, DNA Genotek). Individuals were requested to avoid food intake 30 min prior to the collection and were asked to fill the collection tube to the best of their capacity. Approximately 3 ml of total material (including stabilizing liquid) was collected from individuals. Samples were transported to the NCIG lab in checked-in baggage in flight at the end of community visits. Saliva tubes were subject to large changes in temperature during collection and transport, and were kept at room temperature or at 4 °C until further processing. Samples were split into 2 or more aliquots of 1 ml each depending on the quantity of material available after heating samples at 50 °C for 2 h, as recommended by the manufacturer. Each sample tube was separately processed in a hood to reduce handling errors, cross contamination and external contamination. One of the aliquots with 1 ml sample was used for the DNA extraction and remaining aliquots were stored at −20 °C or −80 °C for long term storage.

DNA extractions from saliva were performed by Australian Phenomics Facility (APF) on QIAsymphony SP using QIAsymphony DSP DNA Midi Kit (QIAGEN). In brief, tubes were incubated at 56 °C for 1 h followed by addition of 2 µl of RNAse A (100 mg ml$^{-1}$, QIAGEN) to 1 ml of saliva sample and incubated at room temperature for 5 min. RNAse activity was suppressed by incubation at 50 °C for 40 min. A custom protocol on QIAsymphony SP, specifically developed for 1 ml of saliva sample was then used to run the DNA extraction process on the instrument. DNA was eluted in 100 µl TE buffer.

### Fresh blood collection and DNA extraction

We collected fresh blood where possible from consenting individuals in BD Vacutainer EDTA tubes (lavender caps, BD). Blood tubes were immediately placed on ice after the collection and shipped to the NCIG laboratory on dry ice. Blood samples were stored at −80 °C until required. DNA extraction was performed using FlexiGene DNA Kit (QIAGEN) according to the manufacturer's protocol. In brief, frozen blood samples were thawed in 37 °C water bath and mixed with the lysis buffer. A cell pellet was then collected by centrifugation at 2,000$g$ for 5 min and supernatant was discarded. Cell pellet was mixed with denaturation buffer and protease enzyme followed by incubation at 65 °C in water bath for 10 min. DNA was precipitated using isopropanol and centrifuged at 2,000$g$ for 3 min. DNA pellet was then washed with 70% ethanol and pelleted at 2,000$g$ for 3 min. Finally, supernatant was discarded and the DNA pellet was air dried. Dry DNA pellet was resuspended in 1 ml of hydration buffer (10 mM Tris-Cl) and incubated at 65 °C for 1 h for dissolution.

For non-Indigenous samples, HMW DNA was previously extracted from blood, using Qiagen DNeasy Blood and Tissue Kit, as per manufacturer's instructions, and stored at −80 °C.

### Whole-genome ONT sequencing

HMW DNA samples were transferred to the Garvan Institute Sequencing Platform for long-read sequencing analysis on Oxford Nanopore Technologies (ONT) instruments. DNA quantity was measured using a Qubit (Thermo Fisher Scientific), purity on a NanoDrop (Thermo Fisher Scientific) and fragment-size distribution on a TapeStation (Agilent). Prior to ONT library preparations, DNA was sheared to ~15–20 kb fragment size using Covaris G-tubes. No shearing was performed on samples where the starting fragment distribution peaked at or below ~25 kb. Sequencing libraries were prepared from ~1–2ug of DNA, using native library preparation kits (either SQK-LSK110 or SQK-LSK114), according to the manufacturer's instructions. Each library was loaded onto a PromethION flow cell (R9.4.1 for SQK-LSK110 libraries, R10.4.1 for SQK-LSK114 libraries) and sequenced on an ONT PromethION P48 device. Samples were run for a maximum duration of 72 h, with 1–3 nuclease flushes and reloads performed during the run, where necessary to maximize sequencing yield.

### ONT data processing

Raw ONT sequencing data were converted from FAST5 to the more compact BLOW5 format[43] in real-time on the PromethION during each sequencing run using slow5tools[44] (v.0.3.0). BLOW5 data were transferred to the Australian National Computational Infrastructure (NCI) high-performance computing environment before further processing. Data were base-called with Guppy (v.6.0.1), using the Buttery-eel wrapper for BLOW5 input[45], with the high-accuracy model and reads with mean quality <7 were excluded from further analysis.

### Alignment to reference genome

To evaluate the use of hg38 and T2T-chm13 reference genomes, ONT libraries generated in our study for the HG001 and HG002 reference samples and matched Illumina libraries from the GIAB consortium were mapped against each reference genome. The short-read data were mapped using bwa-mem2 (v.2.2.1), with -$Y$ optional parameter, and the long-read data were mapped using minimap2[46] (v.2.22) with the following optional parameters: -x map-ont -a –secondary=no –MD. The alignment of each individual library to either hg38 or T2T-chm13 was made in a sex-specific manner with an XY reference for genotypically male individuals and an XO reference for genotypically female individuals. After selecting T2T-chm13 as our central reference genome, all other ONT libraries in the cohort were also mapped to this reference using minimap2, as just described.

### Detection of non-human reads

To assess the impact of microbial contamination in our sequenced libraries, we first used Centrifuge[47] to identify and classify all non-human reads. We then measured the rate of alignment for these reads to the chm13 reference genome within our standard workflow (see above). We found negligible erroneous alignment of non-human reads to human chromosomes, thereby mitigating the risk of microbial reads causing detection of erroneous variants (see Extended Data Fig. 1a,b).

### Detection of structural variation

Detection of large indels (20–49 bp) and SVs (≥50 bp) on HG001 and HG002 Illumina mapped libraries was performed using smoove (v.0.2.6) with default parameters. Variant detection with ONT mapped libraries was performed on each individual sample using *CuteSV*[24] (v.1.0.13) with the following optional parameters: --max_cluster_bias_INS 100 --diff_ratio_merging_INS 0.3 --max_cluster_bias_DEL 100 --diff_ratio_merging_DEL 0.3 --report_readid --min_support 5 --min_size 20 --max_size 1000000 --genotype. Deletions with <20% of supporting reads for the variant sequence were excluded and insertions with <5% of supporting reads were also excluded. Individual callsets were then merged into a unified joint-call catalogue using Jasmine[27] with the following optional parameters: --min_support = 1 --mark_specific spec_reads= 7 spec_len=20 --pre_normalize --output_genotypes --allow_intrasample–clique_merging --dup_to_ins --normalize_type --run_iris iris_args=min_ins_length=20, --rerunracon, --keep_long_variants. Variants in the joint

non-redundant callset were filtered to exclude events with weak evidence (QUAL ≤ 5).

## Benchmarking against HGSVC_v4 HG002

We assessed our SV detection strategy using ONT data from the HG002 reference individual, using data from both LSK110/R9.4.1 and LSK114/R10.4.1 ONT chemistry. To determine the average coverage of each library, we computed the total number of bases across all aligned reads and divided it by the genome size. Subsequently, we conducted downsampling by randomly selecting reads until reaching the target coverage. This downsampling process was repeated iteratively to achieve coverages ranging from 30X down to 5X for both datasets.

To evaluate SV detection accuracy, we compared our results to the HGSVC (Freeze 4) annotation for HG002, which served as the 'truth set'. SVs were deemed true positives (TP) if they matched in type, exhibited a minimum of 50% reciprocal overlap, and had breakpoints within a 200 bp range of each other. SVs in our dataset that did not meet these criteria were classified as false positives (FP), while SVs present in the truth set without a corresponding match were considered false negatives (FN). We used precision and recall metrics to assess the performance of our SV detection method at different coverage levels. Precision was calculated as TP/(TP + FP). Recall was calculated as TP/(TP + FN). Results are presented in Extended Data Fig. 2a.

## Structural variation repeat classification

Indels and SVs were classified according to repeat type using custom analysis methods. We first created an extended local allele sequence for each variant, which was 5× the size of the variant itself. For each insertion, we created an extended ALT allele by extracting reference sequence from the T2T-chm13 genome immediately upstream (2× variant size) and downstream (2× variant size) of the variant site, then concatenating these in appropriate order with the consensus insertion sequence that was retrieved from the Jasmine VCF. For each deletion, we created an extended REF allele by extending the variant position in either direction (by 2× variant size) and extracting reference sequence from the T2T-chm13 genome.

Each extended allele, which captures the variant in its local sequence context, was then scanned for tandem repeats using Tandem Repeat Finder[48] (trf409.linux64) with input parameters recommended by the developers (2 7 7 80 10 50 500). Annotated tandem repeats were parsed by their period: 1 bp, homopolymer (HOMO); 2–12 bp, STR; >12 bp, TR. Any overlapping annotations of the same type were merged. We then calculated the extent to which the variant site (that is, central 20% of the local sequence allele) was covered by repeats of each type; if ≥75% of the variant was covered by repeats of a single type, the variant was classified accordingly as either HOMO, STR or TR.

Each extended allele was then scanned for interspersed mobile elements using RepeatMasker (4.1.2-p1) with the following input parameters: -species human -gff -s -norna -nolow. Annotated interspersed repeats were parsed into different types (SINE, LINE, DNA transposon, LTR, retroposon or other), based on RepeatMasker classifications, and labelled as 'complete' (≥75%) or 'fragment' (<75%) based on the fraction of the canonical sequence element that was present. We then calculated the extent to which the variant site within its local allele sequence was covered by interspersed repeats; if ≥75% of the variant was covered by an element or elements of a single type, the variant was classified accordingly as either: SINE, LINE, DNA transposon, LTR, retroposon or other. If the variant itself covered at least ≥75% of one or more complete annotated elements, the variant was labelled as a 'complete' transposition event. If not, it was labelled as a mobile element 'fragment', which are mostly small SVs contained within larger interspersed elements. Variants that were not classified with either a tandem or interspersed repeat label, were considered 'non-repetitive'.

We detected 6,947 (2.3%) homopolymeric deletions–insertions, although we note these are likely to be enriched for technical errors, based on known ONT sequencing error profiles[49], and they were therefore excluded from subsequent analyses.

## Comparison to annotations

To assess the novelty of our SV catalogue, we compared SVs to three reference datasets: (1) the gnomAD (v.2.1) SV database (http://ncbi.nlm.nih.gov/sites/dbvarapp/studies/nstd166/); (2) an SV callset from population-scale ONT sequencing of Icelanders published recently by deCODE genetics (http://github.com/DecodeGenetics/LRS_SV_sets); (3) and the HGSVC (freeze 4; http://ftp.1000genomes.ebi.ac.uk/vol1/ftp/data_collections/HGSVC2). To ensure comparability, SV coordinates were first converted from T2T-chm13 to the hg38 reference genome using the LiftOver utility from UCSC. Successfully lifted SVs were then intersected with gnomAD, deCODE and HGSVC annotations, separately. To account for variability in methods of SV detection between the different studies, as well as potential confounding effects during LiftOver, we allowed for some discrepancy in the placement of breakpoints; SVs were classified as having high (>80%), moderate (50–80%), low (<50%) or no (0%) reciprocal overlap to an annotated SV in gnomAD, deCODE and/or HGSVC. We considered SVs with high and moderate reciprocal overlap as 'annotated' and SVs with low or no reciprocal overlap as not 'unannotated', and therefore potentially novel.

## Telomere distance

The distance of each variant to the nearest telomere was calculated and based on that distance variants were binned into 500 kb fixed windows. The number of variants in each bin was counted and averaged across all chromosomes to assess the density of variant distribution across a generic chromosome. There was a higher density of variants within the 5 Mb of the telomere region, including acrocentric and metacentric chromosomes. To investigate the types of variants driving this effect, we parsed the variant counts in the 500-kb bins based on variant type. We then fitted a LOESS curve onto the log-transformed counts of the different bins for each variant type, displaying the results in an exponential scale, to ensure that $y$ values in the regression were all greater than 0. The curves for different variant types showed that tandem repeats were the variants mostly driving the higher density near telomeres.

## PCOA and genetic distance

The non-redundant variant callset was converted into a binary matrix, with rows representing variants and columns representing individuals. The presence of a variant in an individual was represented with a 1 and the absence with a 0. We then used the vegdist function from the vegan R package to calculate the dissimilarity between individuals based on their variant composition using Bray–Curtis methodology. Bray–Curtis was chosen for: (1) its ability to distinguish closely related individuals; (2) its robustness to experimental variables (for example, variable coverage) that may result in missing data; and (3) its previously demonstrated usage for analysis of SVs[20].

Subsequently, we performed a PCOA on the dissimilarity matrix using the pcoa function from the ape R package. PCOA1 and PCOA2, representing principal coordinate axes 1 and 2, were plotted for each individual according to their community/group. We calculated the percent of variance explained by PCOA1 and PCOA2 by dividing their corresponding eigenvalues to the total sum of eigenvalues for all axes multiplied by 100.

To calculate pairwise $F_{ST}$, we randomly selected 10,000 SVs with a frequency higher than 10% in our cohort. This data was converted into a matrix, where rows represented individuals, the first column indicated the group (NCIG-P1, NCIG-P2, NCIG-P3, NCIG-P4 and non-NCIG) and subsequent columns represented loci. The pairwise $F_{ST}$ values between groups was then calculated using the gene.dist function from the hierfstat R package, employing the Weir and Cockerham method[50] (method = WC84).

## Discovery curves

To measure SV diversity, we generated discovery curves, wherein we calculate the number of new non-redundant SVs gained as additional individuals are considered. Starting with a single NCIG individual, the number of non-redundant variants was calculated each time a new individual was added to the analysis, until all 141 individuals were included. The growth rate of the non-redundant set declines as the number of cumulative individuals increases. We then used the values obtained in the discovery curve to generate a log regression model of the number of non-redundant variants as a function of the number of individuals sampled with the lm function from the stats R package. The curves model the level of heterogeneity in a given group, and enable estimation of the number of individuals required to saturate variant discovery. We then generated discovery curves and log regression models by parsing the variants for each community/group (NCIG-P1, NCIG-P2, NCIG-P3, NCIG-P4, non-NCIG and NCIG-P1/NCIG-P2/NCIG-P3 combined), according to geographical distribution (NCIG-only, NCIG-absent, global) and variant type (non-repetitive, tandem repeats and mobile elements).

## LOEUF constraint analysis

We binned variants in the non-redundant callset intersecting protein-coding genes based on their LOEUF decile, previously assigned by gnomAD, and which measures intolerance to loss-of-function variation. Genes in the 1st decile have the highest constraint, while genes in the 10th decile are the least constrained. Therefore, if variants (large indels and/or SVs) regularly have deleterious effects on gene function, the expectation would be that genes in the 1st decile would harbour relatively fewer variants than genes in higher deciles, after accounting for gene size. To test this, we calculated the variant density within CDS and non-CDS regions (introns, UTRs and ±2 kb flanking regulatory regions) of all the genes in each LOEUF decile. Density was calculated by counting the number of non-redundant variants intersecting CDS regions of all genes in a given decile, divided by the total size of CDS regions of all genes of that decile. Similarly, we counted the number of variants intersecting non-CDS regions (but not intersecting CDS regions) and divided that by the total size of non-CDS regions of all the genes in a given decile. Variant density was plotted per LOEUF decile for CDS & non-CDS regions, showing clear differences between high and low deciles for both CDS and non-CDS regions.

## Analysis of STRs

To explore the landscape of STR variation, we retrieved all joint-called indels and SVs classified above as 'STR' variants, which represent expansions (that is, insertions) and contractions (that is, deletions) of local STR elements. For each variant, we recorded the total expansion/contraction size, the period size (2–12 bp) and the STR motif identified by Tandem Repeat Finder (see 'Structural variation repeat classification' above), and investigated global frequencies for each of these dimensions. For STR motif frequency analysis, we considered all possible motif representations in both orientations as a single redundant motif (for example, CAG, AGC, GCA, TGC, GCT, CTG are a single redundant triplet). To identify significantly expanded STR sites, we applied the following criteria: (1) STR period size ≥3 bp; (2) local STR element expanded by at least ≥10 repeats; (3) local STR element is expanded by ≥50% of its reference size; and (4) local STR element reference size is <1 kb. This identified 651 STR sites within protein-coding genes that were significantly expanded in at least one individual.

Individual-level, diploid genotyping of STR alleles was used to elucidate full allelic diversity at the 651 STR sites within protein-coding genes that were significantly expanded in at least one individual (see above), as well as 50 known disease-associated STR loci. This was performed using a custom analysis method. In brief, we identified variation within the local region around a given STR site using clair3[51] (v.0.1-r12; for

SNVs and 2–20 bp indels) and sniffles2[52] (v.2.0.2; for indels/SVs >20 bp). Sniffles2 was used instead of CuteSV (as above) because it shows better performance at STR sites when guided by the --tandem-repeats input parameter. Variants from clair3 and sniffles2 were incorporated in a haplotype-specific fashion into the local genome sequence using bcftools consensus (v.1.12)[53], and the modified hap1/hap2 sequences were extracted in a ±50-bp window centred on the STR site; these constitute the consensus STR allele sequences for a given individual at a given STR site, with the larger being designated 'allele_A' and the shorter 'allele_B'. Tandem Repeat Finder was used to determine the STR period size, length, motif and other summary statistics for each STR allele. Allele sequences were visualized in sequence bar charts, in which each tile represented a nucleotide (A, C, G and T), using R package ggplot2. To investigate the variability of STRs within the different communities, we calculated the mean and standard deviation of STR lengths for the alleles within each community. We then performed an ANOVA test ($P < 0.05$) to identify STR sites that were significantly variable between communities. We plotted the standard deviation for each significantly variable site, normalized to range between 0 and 1, as a heatmap and also performed hierarchical clustering using the heatmap.2 function of the gplots R package.

## Analysis of large CNVs

We detected large CNVs (>50 kb) in individual libraries using CNVpytor (v.1.3.1) with a 10 kb bin size. To maintain only high-confidence predicted CNVs, we excluded calls with a $P$ value $> 10^{-4}$. We grouped CNVs with more than 50% reciprocal overlap into merged regions of variable copy number, encompassing all the individual calls contained in that region. We counted both the number of CNVs identified in each individual and the number of individuals with a call within a given CNV region. That way we classified CNV regions according to the number of different individuals with a call within that CNV region (singleton: 1 individual; polymorphic: >2 and <50% of individuals; major: ≥50% of individuals and <all individuals) and also range depending on its distribution across NCIG and non-NCIG groups (NCIG-only: only found among NCIG individuals; NCIG-absent: only found among non-NCIG individuals; global: found among both NCIG and non-NCIG individuals). To identify annotated protein-coding genes within CNV regions, we used bedtools intersect with a requirement for complete gene containment within a given region (parameter -f 1.0). Read-depth visualization for multiple samples at a specific CNV region was achieved by extracting values from bigwig files and normalizing based on library size.

## Data analysis

All data manipulation and visualization, as well as plotting was performed in R (v.4.0.0).

## Reporting summary

Further information on research design is available in the Nature Portfolio Reporting Summary linked to this article.

## Data availability

The following publicly accessible datasets were used in this study: (1) the gnomAD (v.2.1) SV database: http://ncbi.nlm.nih.gov/sites/dbvarapp/studies/nstd166/; (2) deCODE genetics SV callset: http://github.com/DecodeGenetics/LRS_SV_sets; and (3) HGSVC (freeze 4): http://ftp.1000genomes.ebi.ac.uk/vol1/ftp/data_collections/HGSVC2. The following reference genomes were used in this study: T2T-chm13 (v.2.0): https://github.com/marbl/CHM13 and Hg38 (GRCh38.p13): https://www.ncbi.nlm.nih.gov/datasets/genome/GCF_000001405.39/. All raw sequencing data, processed output files and associated metadata are permanently stored on Australia's National Computational Infrastructure (NCI) under the control of the Collection Access and

Research Advisory Committee (CARAC) appointed and overseen by the NCIG Indigenous-majority governance board. Requests for access by external researchers will be considered by CARAC and governed by the NCIG Board. Data access requests from external researchers may be granted when the board is satisfied that core principles of Indigenous engagement are observed within the proposed research. At the heart of this is the requirement that the proposed research will be of benefit to Australian Indigenous peoples and is identified as important by the communities whose data is involved. Further information can be found within the NCIG governance framework (https://ncig.anu.edu.au/files/NCIG-Governance-Framework.pdf). Data access requests should be directed to jcsmr.ncig@anu.edu.au.

## Code availability

All original code has been deposited at Zenodo and is publicly available from https://doi.org/10.5281/zenodo.10020534.

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

**Acknowledgements** We acknowledge the Aboriginal and Torres Strait Islander peoples as the first peoples and traditional custodians of the lands and waters where we meet, live, learn and work. We celebrate the rich diversity of Aboriginal and Torres Strait Islander cultures and the ongoing leadership of our First Nations' peoples and communities who have paved the way. We pay our respects to ancestors of this country, the legacy of elders, the knowledge holders, and leaders of the past, present, and future. This work was conducted primarily on land traditionally owned by the Ngunnawal and Ngambri peoples and the Gadigal people of the Eora Nation. We are indebted to the individuals and communities who participated in this research and the NCIG Governance Board who guided this work in a culturally appropriate manner. We acknowledge the following Community Organizations and individuals: Yarrabah Shire Council, R. Andrews, E. Fourmile and P. Burns; Tiwi Land Council Board members; Yalu Aboriginal Corporation (Galiwin'ku), R. Wunungmurra, E. Djotja, R. Gundjarrangbuy; Titjikala Shire Council and Titjikala Health services. We thank our colleagues A. McCarthy, W. Hoy, S. Foote, J. Matthews, R. Thomson, D. MacArthur, J. Yuan, T. Amos and J. Craig for their various contributions to the project. This project was undertaken with the assistance of resources and services from the National Computational Infrastructure (NCI), which is supported by the Australian Government and the Australian National University (ANU). We acknowledge the following facilities that were used during this study: the Garvan Sequencing Platform (GSP) and the Australian Phenomics Facility (APF). We acknowledge the following funding sources: Medical Research Futures Fund (MRFF) grants 2016008, 1173594, 2023126 and 2016124, and National Health and Medical Research Council (NHMRC) grants 2011277 and 2021172.

**Author contributions** I.W.D. and H.R.P. conceived the study, with the support of G.J.M., A.H., B.J.M., M.R.J. and S.E. A.B., A.H., M.R.J. and the NCIG provided Indigenous leadership and oversight to the project. The NCIG provided access to DNA samples from Indigenous communities. O.M.S. and A.W.H. provided access to non-Indigenous comparison data. M.R., J.M.H., I.S., M.A.K. and D.S.B.D. processed samples and performed sequencing experiments. A.L.M.R., H.G., S.R.C., H.R.P. and I.W.D. performed bioinformatics analysis. A.L.M.R. and I.W.D. generated the Figures. A.L.M.R., H.R.P and I.W.D. wrote the manuscript, with input from all co-authors.

**Competing interests** This project receives partial in-kind support from Oxford Nanopore Technologies (ONT) under an ongoing collaboration agreement. A.L.M.R., J.M.H., H.G., H.R.P. and I.W.D. have previously received travel and accommodation expenses from ONT to speak at conferences. H.G. and I.W.D. have paid consultant roles with Sequin Pty Ltd. O.M.S. and A.W.H. hold equity in Seonix Pty Ltd. The authors declare no other competing interests.

**Additional information**
**Correspondence and requests for materials** should be addressed to Hardip R. Patel or Ira W. Deveson.

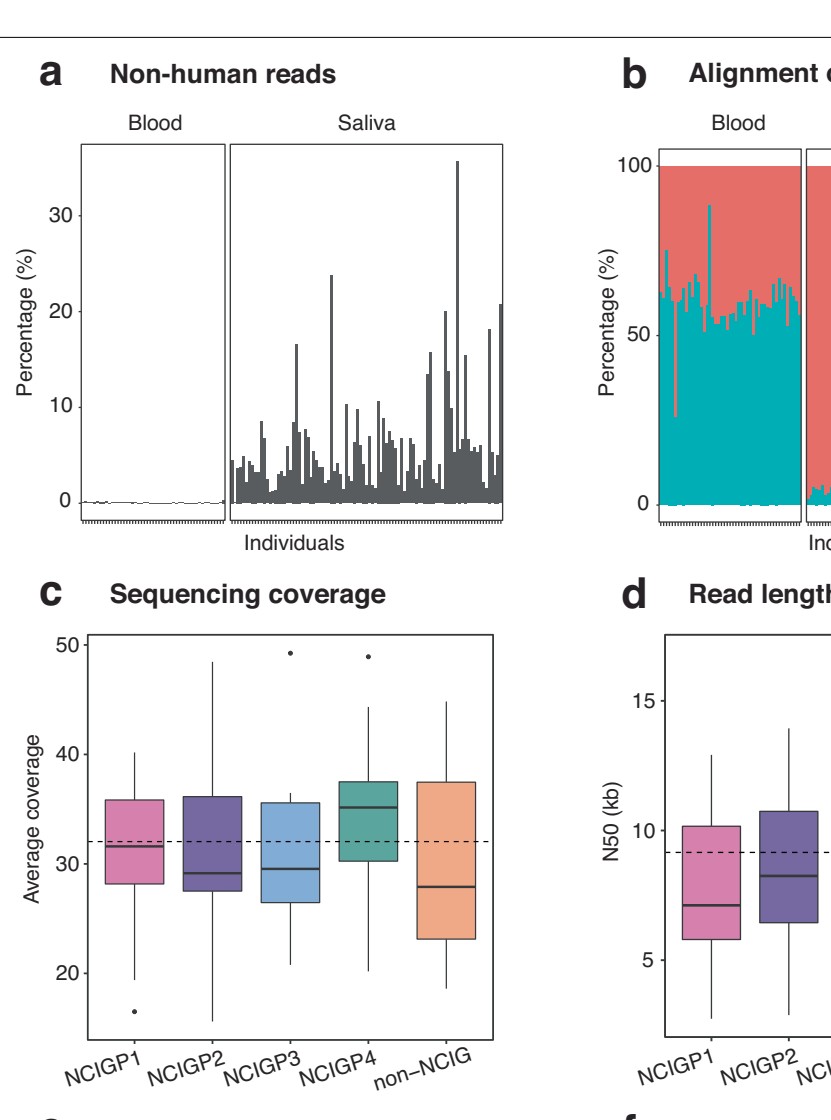

**a** Non-human reads

**b** Alignment of non-human reads

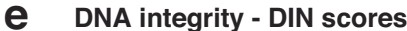
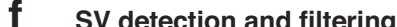

**c** Sequencing coverage

**d** Read lengths

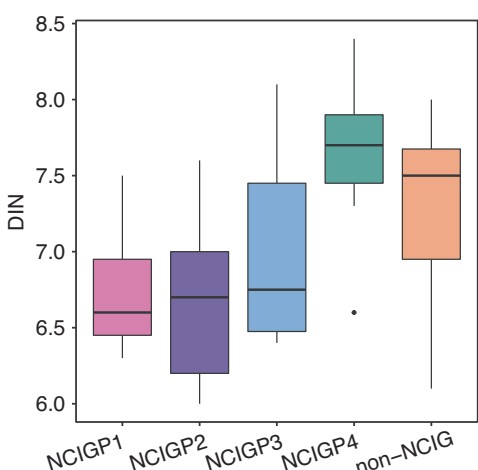

**e** DNA integrity - DIN scores

**f** SV detection and filtering

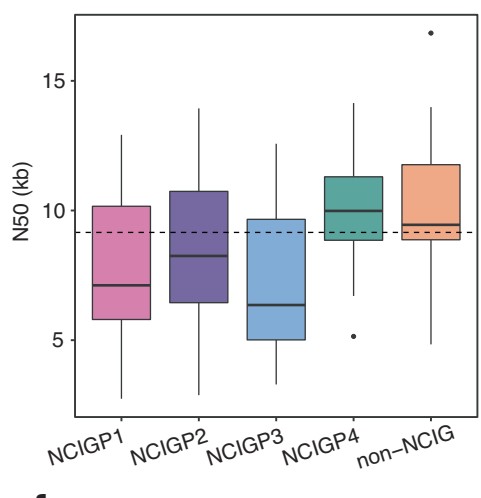
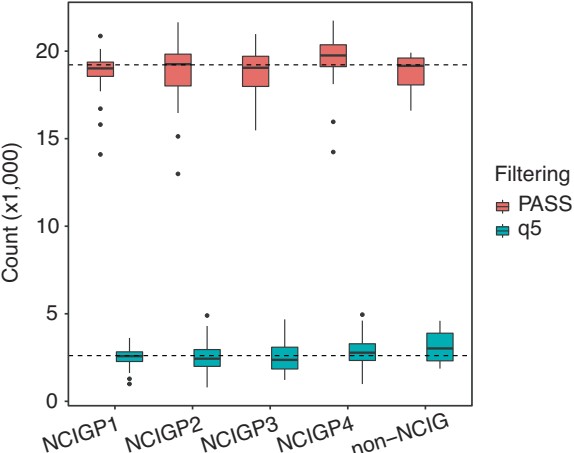

**Extended Data Fig. 1** | See next page for caption.

**Extended Data Fig. 1 | Genomic library characteristics across different groups.** (a) Barchart shows the proportion of non-human reads in sequencing libraries derived from blood or saliva samples. (b) Barchart shows the proportion of mapped (green) and unmapped (orange) non-human reads in sequencing libraries derived from blood or saliva samples. (c) Boxplot shows the average depth of coverage per individual grouped by their communities (NCIGP1 = pink, P2 = purple, P3 = blue, P4 = green & non-NCIG = orange). The horizontal dashed line indicates the average coverage across all libraries in the cohort. (d) Boxplot shows the N50 distribution of individual libraries in the different communities. The horizontal dashed line indicates the average N50 across all libraries in the cohort. (e) Boxplot shows the distribution of DNA Integrity Number (DIN), which indicates the level of fragmentation of a genomic DNA sample, for individual libraries across the different communities. (f) Boxplot shows the distribution of the number of high-quality (PASS=orange) and low-quality (q5=green) structural variants (SVs) per individual grouped by community after quality filtering (Quality ≥ 5). The horizontal dashed lines indicate the average number of high-quality (top) and low-quality (bottom) SVs across all libraries in the cohort. A total of n = 141 individuals (NCIGP1 = 41, NCIGP2 = 32, NCIGP3 = 9, NCIGP4 = 39 and non-NCIG = 20) were examined from independent sequencing experiments in figures c-f. In the boxplots, the middle line is the median, the box represents the interquartile range (IQR), the whiskers extend 1.5 times the IQR from the hinge, and any data points beyond the whiskers are shown individually.

**a** **Benchmark of SV calling pipeline on HG002 sequenced with LSK110 & LSK114 at different coverages against HGSVCv4 reference**

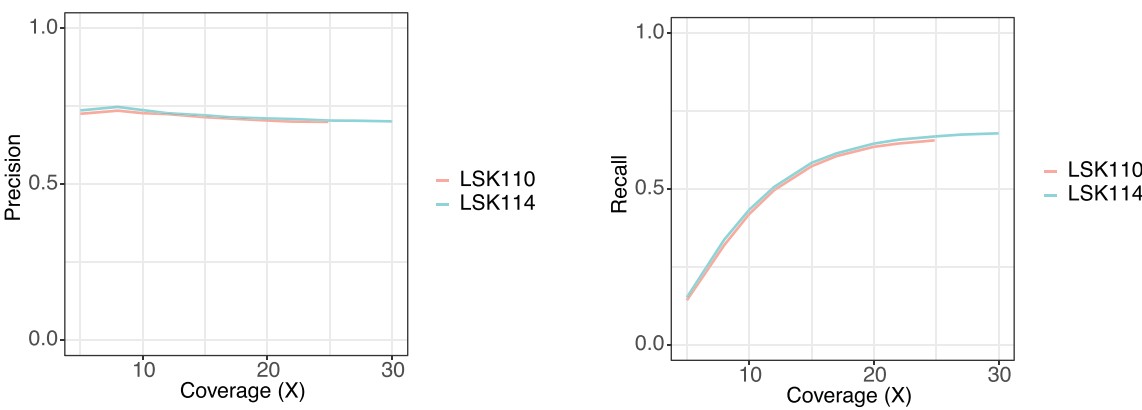

**b** **Repetitive medically-relevant genes, such as *MUC1*, are best resolved by long-read sequencing aligned to the *chm13-T2T* reference.**

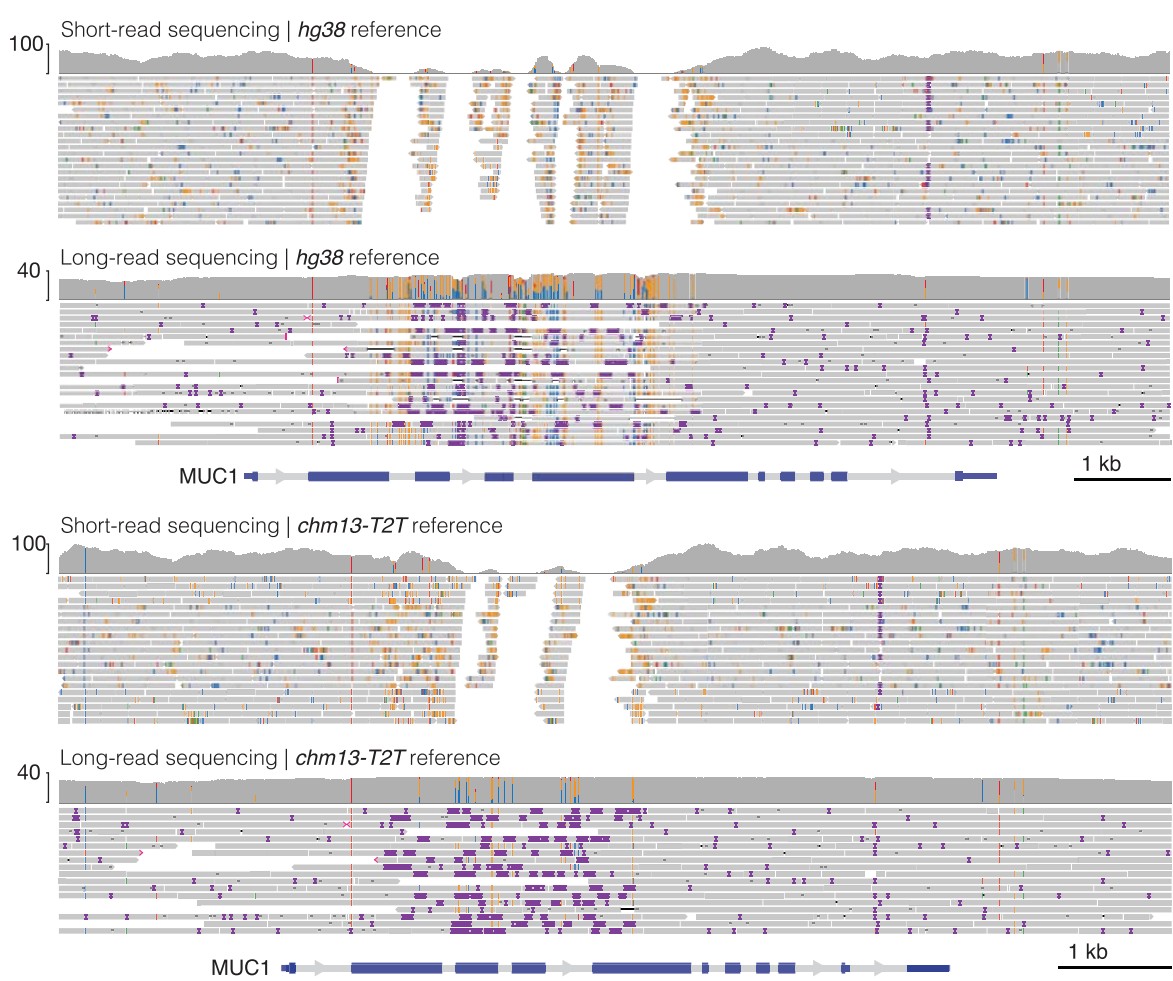

**Extended Data Fig. 2 | Structural variation detection performance and genomic alignments for HG002 samples.** (a) Line plots show precision and recall of SV calls detected with cuteSV for HG002 samples sequenced with LSK110/R9.4.1 or LSK114/R10.4.1 ONT chemistry and subsampled to different coverages compared against HGSVCv4 reference calls for HG002 (taken as a 'truth set'). (b) Genome browser views show comparison of short-read and long-read alignments to either the hg38 or chm13-T2T reference genomes at MUC1, an example of a repetitive medically relevant gene. Both datasets are from the HG002 reference sample. The gene contains a large tandem repeat region that is best resolved by alignment of long-reads to chm13-T2T.

**a**  **Number of CNVs (> 50 kb) per individual and community**

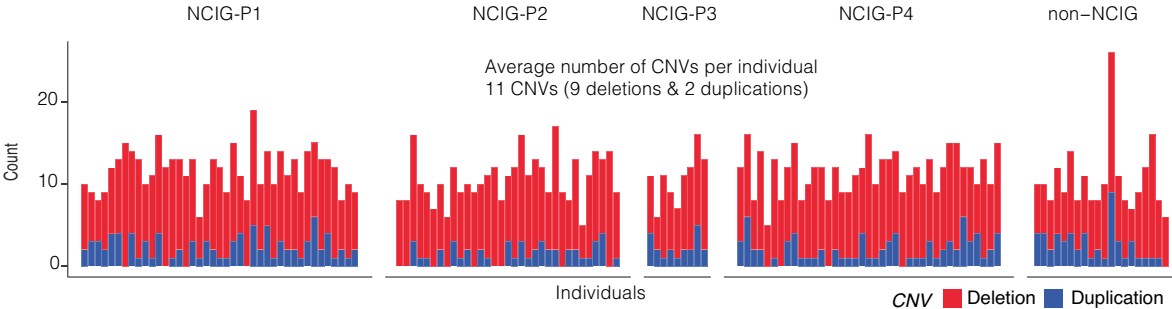

Average number of CNVs per individual
11 CNVs (9 deletions & 2 duplications)

*CNV* ■ Deletion ■ Duplication

**b**  **Cumulative CNV size per individual and community**

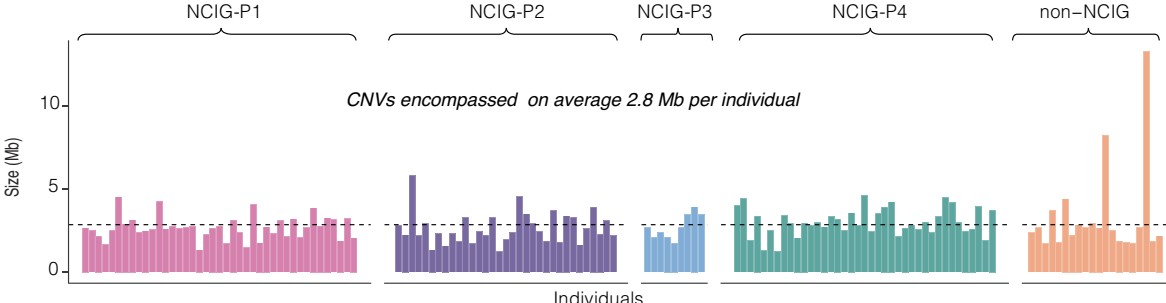

*CNVs encompassed on average 2.8 Mb per individual*

**c**  **Unique CNV regions size distribution**

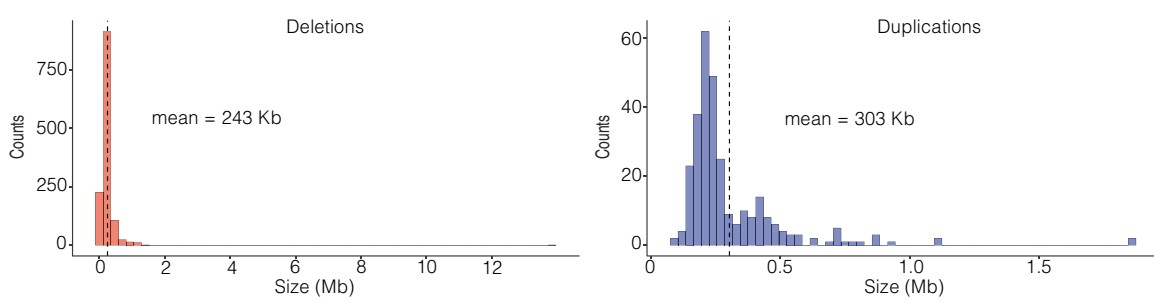

Deletions — mean = 243 Kb

Duplications — mean = 303 Kb

**d**  **Genome Browser view showing coverage tracks for 3 individuals at largest deletion and duplication**

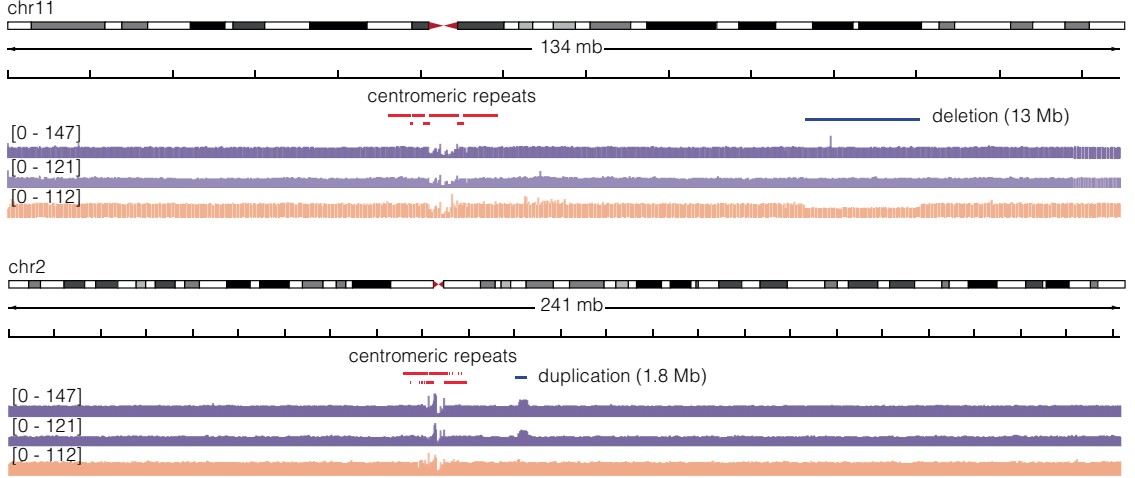

**Extended Data Fig. 3** | See next page for caption.

**Extended Data Fig. 3 | Copy number variation analysis across different groups.** (a) Bar chart shows the number of large CNVs (> 50 kb) identified in individuals from each group, broken down by type: deletion (red) and duplication (blue). (b) Bar chart shows the cumulative size of CNVs identified in individuals from each group. The horizontal dashed line indicates the average cumulative CNV size across the entire cohort. (c) Histograms show the size distribution of unique CNV regions (> 50 kb) containing deletions (red) and duplications (blue). The vertical dashed lines indicate the average size for deletions and duplications, respectively. (d) Genome browser views show coverage tracks for 2 individuals from NCIG-P2 (purple) and 1 non-NCIG individual (orange) across chromosomes 11 and 2 of *chm13-T2T*. In the first panel, the non-NCIG individual has the longest deletion identified, which is indicated by the blue segment and visible in the coverage track for that individual, but missing in the other 2 NCIG-P2 individuals. In the second panel, the 2 NCIG-P2 individuals have the largest duplication identified, also indicated by the blue segment and visible in the respective coverage tracks, but missing in the non-NCIG individual. Centromeric repeats are labeled and represented as red segments.

## a Genome distribution of SVs: parsed by SV-type

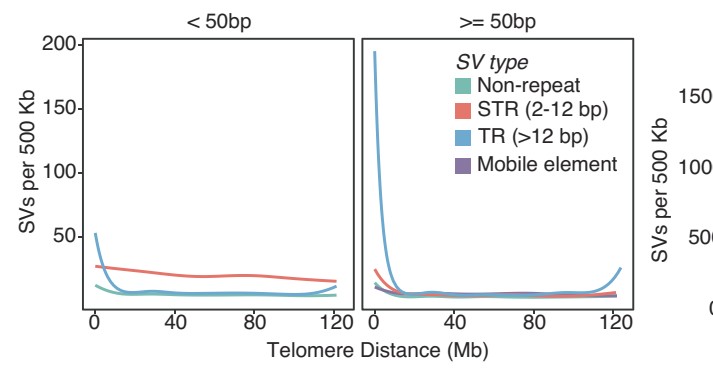

## b Genome distribution of SVs: acrocentric vs non-acrocentric chromosomes

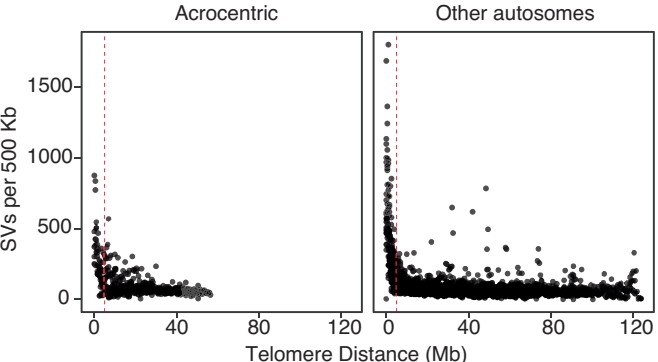

## c Proportion of lifted SVs by type

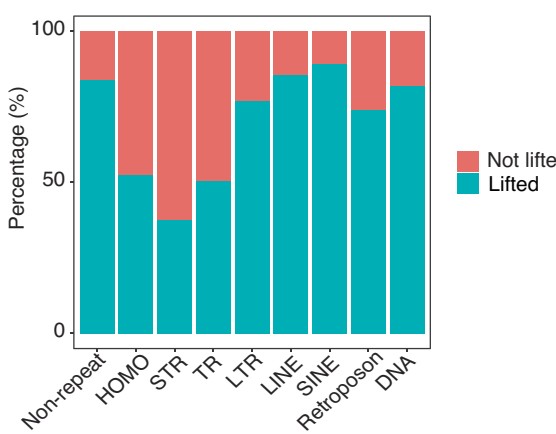

## d Comparison of lifted SVs to existing annotations by SV type

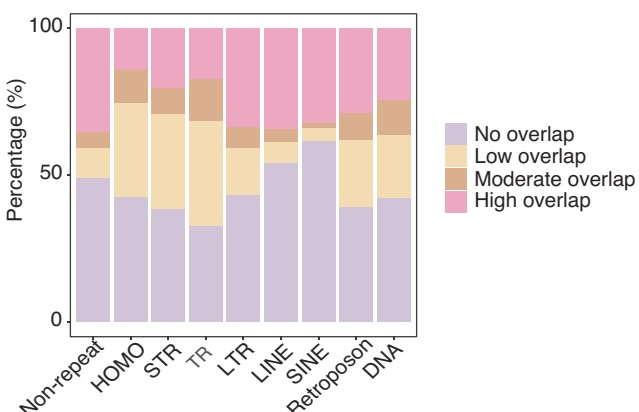

**Extended Data Fig. 4 | Genomic distribution and classification of structural variants.** (a) Line plots show LOESS curves of the number of large indels (> 20 & <50 bp) and structural variants (≥ 50 bp) per 500 Kb fixed window relative to the distance to the nearest telomere, parsed by variant type (non-repetitive = teal, short tandem repeat = red, tandem repeat = blue & mobile element = purple). (b) Dot plots show the number of structural variants per 500 Kb fixed window, relative to the distance of the window to the nearest telomere, for acrocentric and metacentric autosomes. The vertical dashed lines indicate a distance of 5 MB from the telomere. (c) Bar chart shows the proportion of SVs that could be lifted (green) or not (orange) from *T2T-chm13* to *hg38*. (d) Bar chart shows the proportion of SVs classified according to reciprocal overlap as high (>80%; pink), moderate (50-80%; brown), low (<50%; beige) or no overlap (purple) in comparison to reference databases (gnomAD, deCODE & HGSVCv4; see Fig. 2e), parsed by SV type.

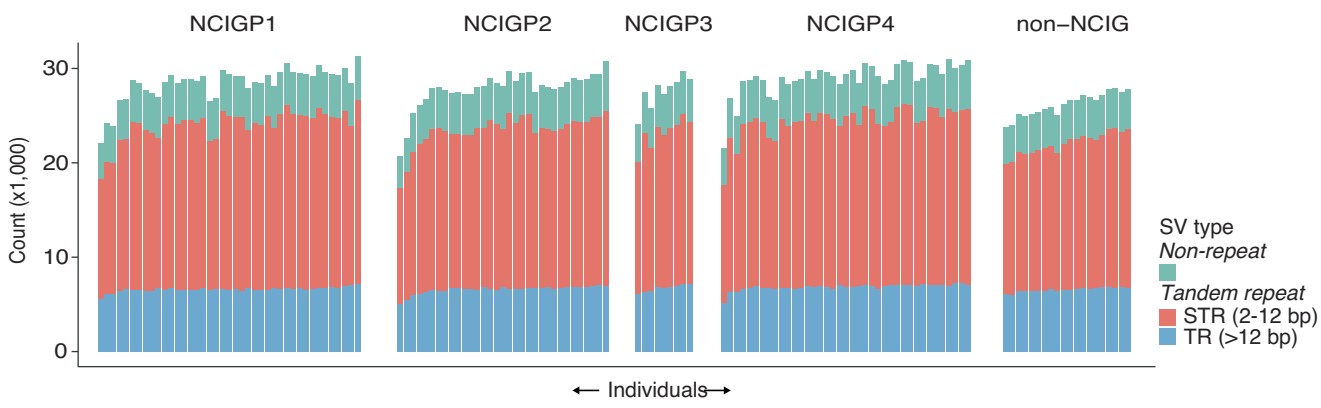

## a  Indels (20-49bp) per individual and community

**SV type**
*Non-repeat* (teal)
*Tandem repeat*
STR (2-12 bp) (red)
TR (>12 bp) (blue)

← Individuals →

## b  Degree of sharedness among indels

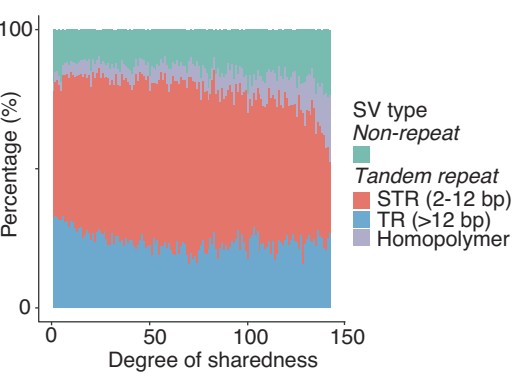

singleton 42534
polymorphic 105127
major 12622
shared 426

## c  Sharedness by indel type

**SV type**
*Non-repeat* (teal)
*Tandem repeat*
STR (2-12 bp) (red)
TR (>12 bp) (blue)
Homopolymer (light purple)

**Extended Data Fig. 5 | Distribution and sharedness of large indels and structural variants across the cohort.** (a) Bar chart shows the number of large indels (20-49 bp) identified in individuals from each group (NCIGP1, P2, P3, P4 & non-NCIG), broken down by type: non-repeat (teal) and tandem repeat (STR = red & TR = blue). (b) Bar plot shows the number of non-redundant structural variants identified within a given number of individuals in the cohort (degree of sharedness). Variants were classified as private (1 individual), polymorphic (> 2 & <50% of individuals), major (≥ 50% of individuals & <all individuals) and shared (all individuals). (c) Bar chart shows the proportion of different variant types (same colour scheme as a, in addition to homopolymers = light purple) for large indels identified within a given number of individuals in the cohort (degree of sharedness).

## a Indigenous-exclusive indels per individual

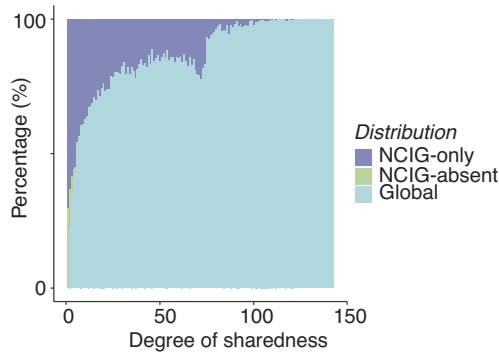

## b Sharedness by indel distribution

## c Novelty of Indigenous vs non-Indigenous SVs

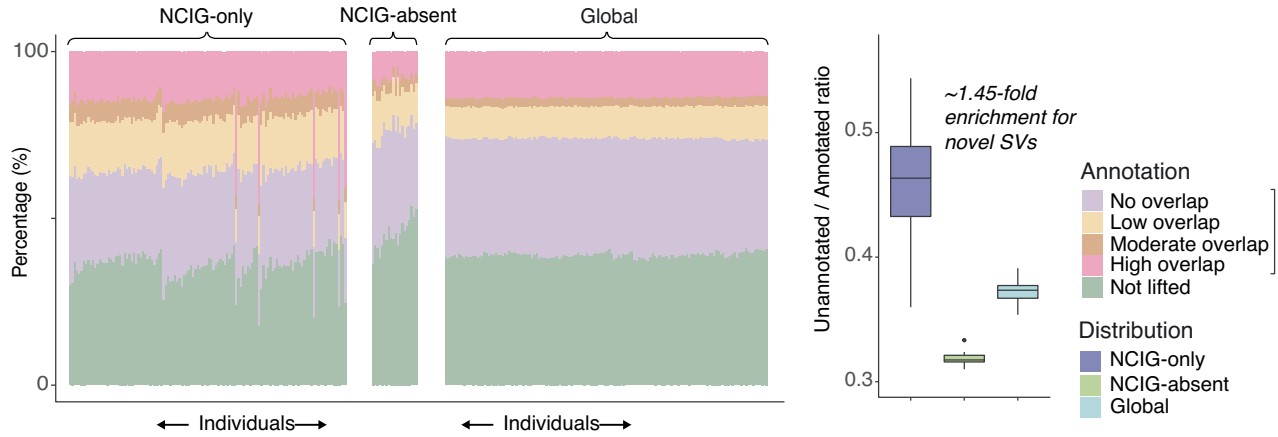

**Extended Data Fig. 6 | Population distribution and annotation of large indels and structural variants.** (a) Bar chart shows the proportion of large indels in each individual that were only found in NCIG individuals (NCIG-only = purple), only found in non-NCIG individuals (NCIG-absent = green) and found in both NCIG & non-NCIG individuals (Global = light blue). (b) Bar chart shows the proportion of NCIG-only, NCIG-absent and Global large indels (same colour scheme as Fig. 3a) for all the variants identified within a given number of individuals in the cohort (degree of sharedness). (c) Proportion of NCIG-only, NCIG-absent or Global SVs in each individual that were previously annotated in

*gnomAD*, *deCODE* or HGSVCv4 SV catalogs (see Fig. 2e). Accompanying boxplot shows the distribution of ratios between the proportion of SVs 'unannotated' (high & moderate overlap) to the proportion of SVs 'annotated' (low and no overlap) against at least one of the databases. A total of n = 141 individuals (NCIG-only = 121, NCIG-absent = 20 and Global = all individuals) were examined from independent sequencing experiments in figure c. In the boxplot, the middle line is the median, the box represents the interquartile range (IQR), the whiskers extend 1.5 times the IQR from the hinge, and any data points beyond the whiskers are shown individually.

## a   Genetic distance between groups

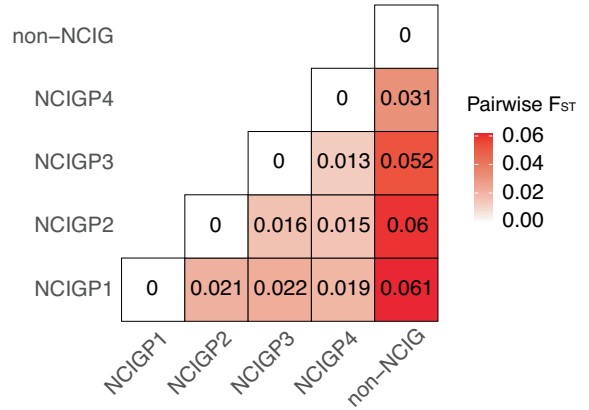

## b   NCIG-only indels (20-49bp) shared between communities

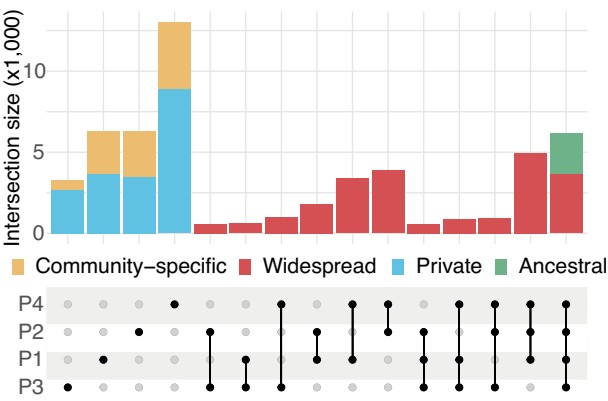

## c   NCIG SNVs shared between communities

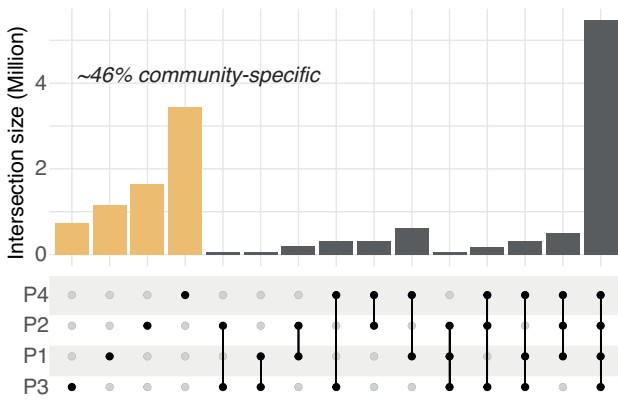

## d   NCIG SNVs comparison to existing annotation

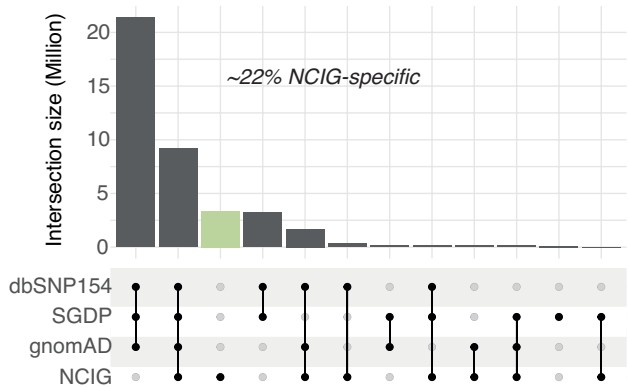

**Extended Data Fig. 7 | Distribution and diversity of genomic variation across different indigenous communities.** (a) Matrix shows pairwise FST calculated using a random set of 10,000 SV loci found among NCIG communities and the non-NCIG group. In the accompanying scale, the intensity of red increases with the FST values. (b) Upset plot shows the distribution of NCIG-only large indels (20-49 bp) shared among the four indigenous communities (NCIGP1, P2, P3 & P4). Variants were classified as private (n = 1 individual; blue), community-specific (n > 1 individual in

1 community; yellow), widespread (n > 1 individual in more than 1 community; red) or shared (n > 1 individual in all 4 communities; green) according to the number of communities in which they were identified. (c) Upset plot shows the distribution of NCIG SNVs shared among the four indigenous communities (NCIGP1, P2, P3 & P4). Community specific variants were highlighted in yellow. (d) Upset plot shows the comparison of NCIG SNVs against the Simon Genome Diversity Project (SGDP), gnomAD and dbSNP154. NCIG specific SNVs were highlighted in green.

## a  Community distributions by SV/indel types

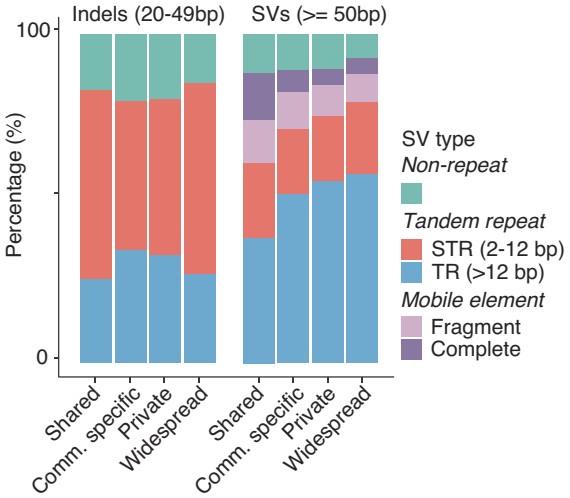

## b  SV diversity

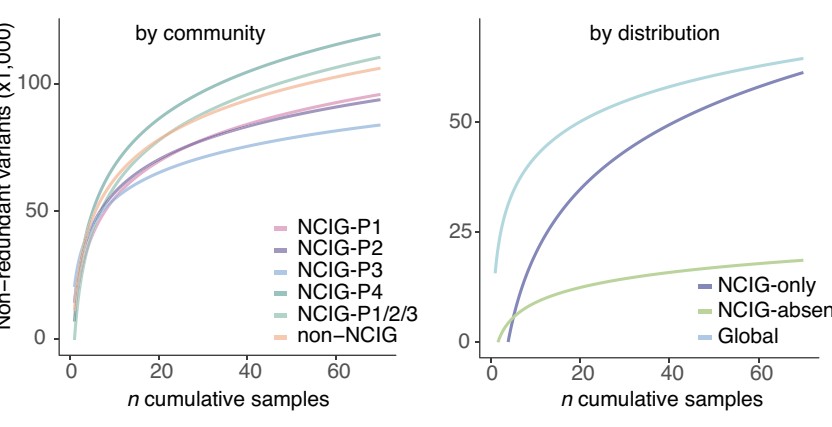
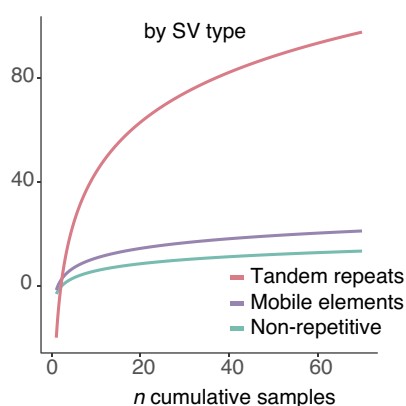

## c  Indel discovery curve

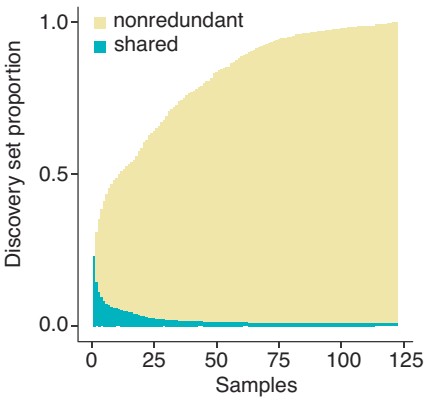

## d  SV diversity by indel type

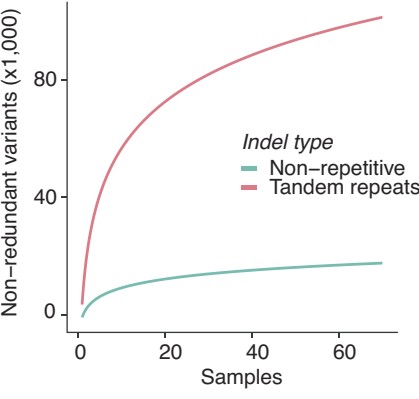

## e  Proportion shared vs exclusive indels

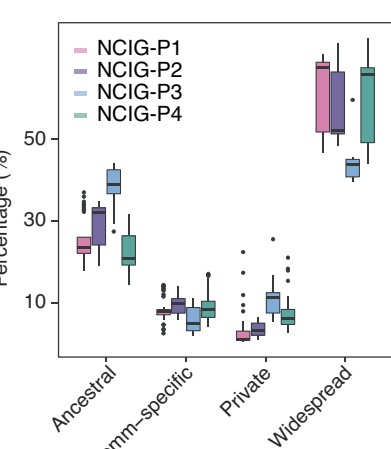

**Extended Data Fig. 8** | See next page for caption.

**Extended Data Fig. 8 | Genomic variation distribution and sampling dynamics across the cohort.** (a) Proportion of different SV types for NCIG-only variants classified as private, community-specific, widespread or shared. Types are non-repetitive (teal), tandem repeat (STR = red & TR = blue) and mobile element (fragment = light purple & complete = dark purple). (b) Log regression models predicting the number of non-redundant SVs identified, given the number of individuals sampled. The models are broken down by community (left panel), by geographical distribution (centre panel) and SV type (NCIG individuals; right panel). (c) Bar chart shows a discovery curve, in which starting with a single NCIG individual, the number of new non-redundant large indels is counted by iteratively adding the unique calls from additional NCIG individuals. Indels shared among all previously added samples are shown as green portions of each bar. The growth rate of the nonredundant set declines as the number of samples increases. (d) Log regression model showing the predicted number of non-redundant large indels identified given the number of individuals sampled. The model was broken down by variant type (Non-repetitive = teal, Tandem repeats = red). (e) Proportion of private, community-specific, widespread & shared NCIG-only variants among individuals, grouped by community. A total of n = 141 individuals (NCIGP1 = 41, NCIGP2 = 32, NCIGP3 = 9, NCIGP4 = 39 and non-NCIG = 20) were examined from independent sequencing experiments in figure e. In the boxplot, the middle line is the median, the box represents the interquartile range (IQR), the whiskers extend 1.5 times the IQR from the hinge, and any data points beyond the whiskers are shown individually.

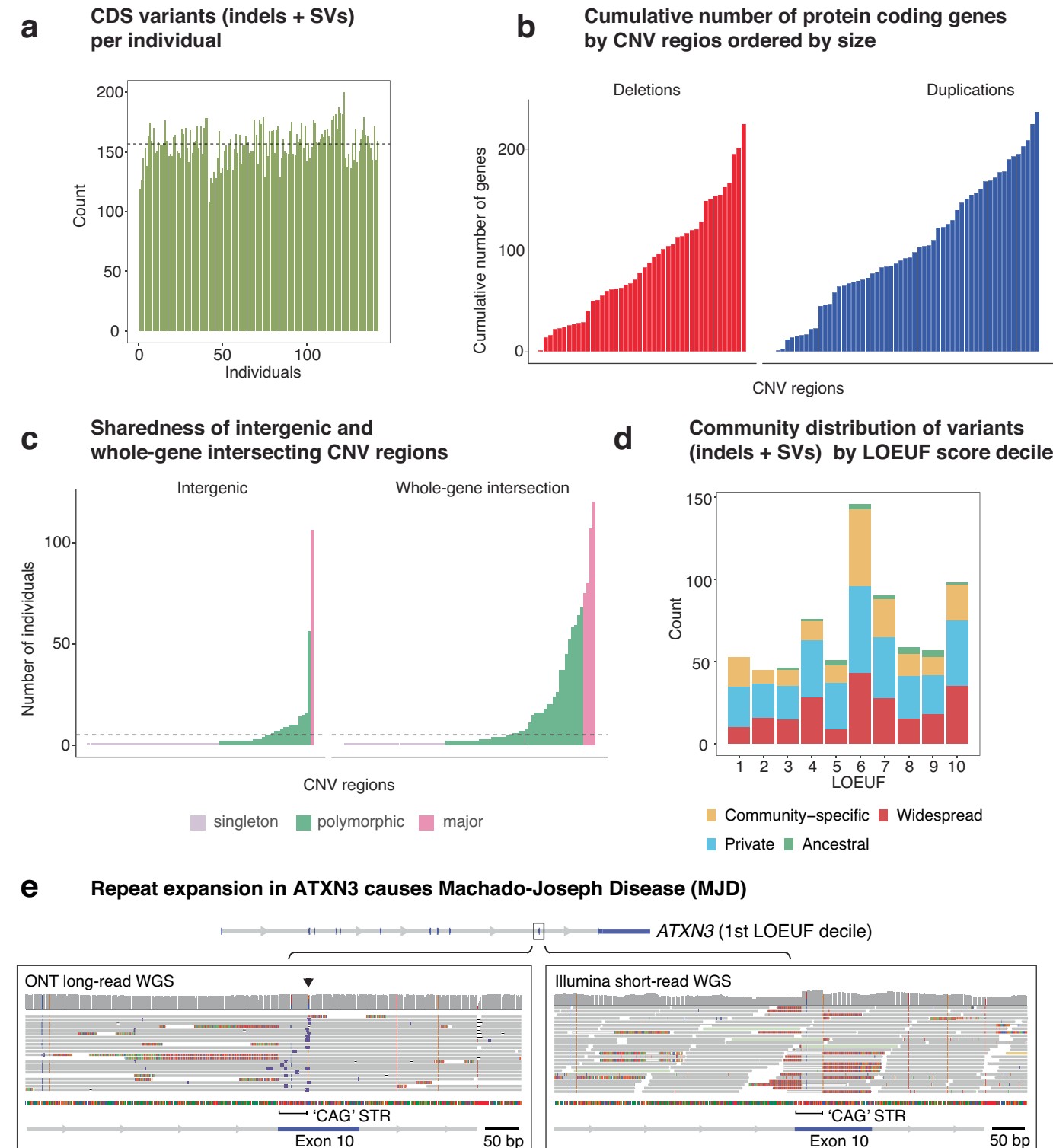

**a  CDS variants (indels + SVs) per individual**

**b  Cumulative number of protein coding genes by CNV regios ordered by size**

Deletions    Duplications

**c  Sharedness of intergenic and whole-gene intersecting CNV regions**

Intergenic    Whole-gene intersection

■ singleton  ■ polymorphic  ■ major

**d  Community distribution of variants (indels + SVs) by LOEUF score decile**

■ Community–specific  ■ Widespread
■ Private  ■ Ancestral

**e  Repeat expansion in ATXN3 causes Machado-Joseph Disease (MJD)**

*ATXN3* (1st LOEUF decile)

ONT long-read WGS    Illumina short-read WGS

'CAG' STR    'CAG' STR
Exon 10    50 bp    Exon 10    50 bp

*130 bp insertion indicates STR expansion*    *Soft-clipped alignments at STR site; SV not resolved*

**Extended Data Fig. 9 | Genomic variation impact on protein-coding genes.**
(a) Bar plot shows the number of variants per individual impacting CDS exons
of protein-coding genes. The horizontal dashed line indicates the average
number of variants in CDS regions across the entire cohort. (b) Bar plots show
the cumulative number of whole-genes contained within CNV regions sorted
by size in increasing order for deletions and duplications. (c) Bar plots show for
each CNV region, the number of different individuals that had a CNV identified
within that region and were either intergenic or intersected whole genes.
Different CNV regions are classified as singleton (light purple; 1 individual),
polymorphic (green; > 2 & < 50% of individuals) and major (pink; ≥ 50% of
individuals & < all individuals). (d) The bar plot shows the number of NCIG-only

variants per LOEUF decile parsed by their level of distribution within NCIG
communities. Variants were classified as private (n = 1 individual; blue),
community-specific (n > 1 individual in 1 community; yellow), widespread (n > 1
individual in more than 1 community; red) or shared (n > 1 individual in all 4
communities; green) according to the number of communities in which they
were identified. (e) Genome browser view shows sequencing alignments to
*ATXN3*. A 'CAG' STR expansion, known to cause Machado-Joseph Disease (MJD),
was identified in one NCIG-P2 individual. ONT reads span the expansion (left
panel; purple markers indicate insertions). Illumina short-reads do not span the
expansion, and are soft-clipped (right panel).

## a  Identifying STR sites in protein coding genes with significant expansions

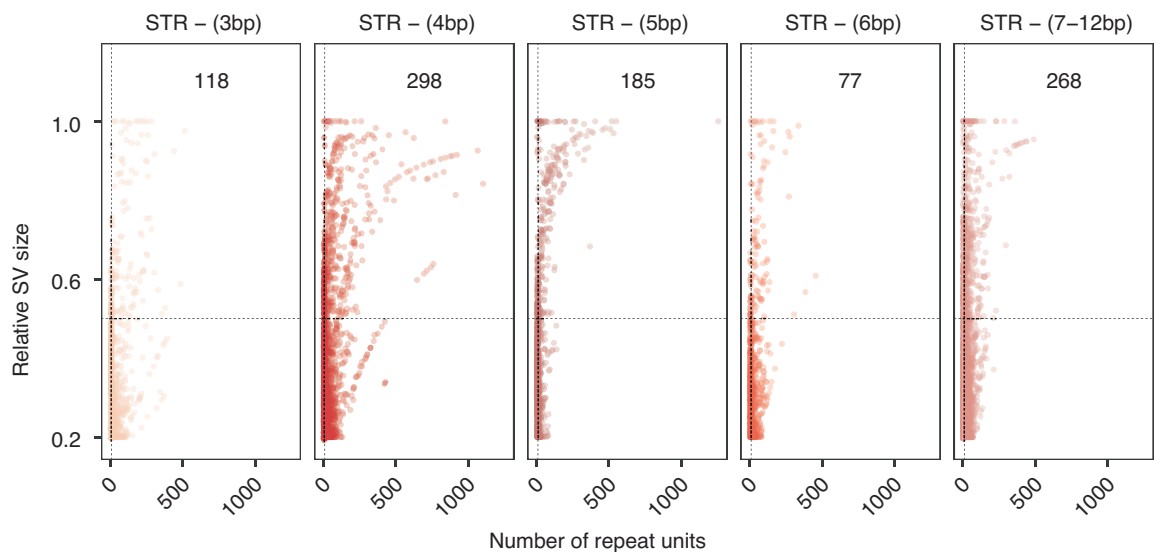

## b  Diversity of STR expansions within protein-coding genes

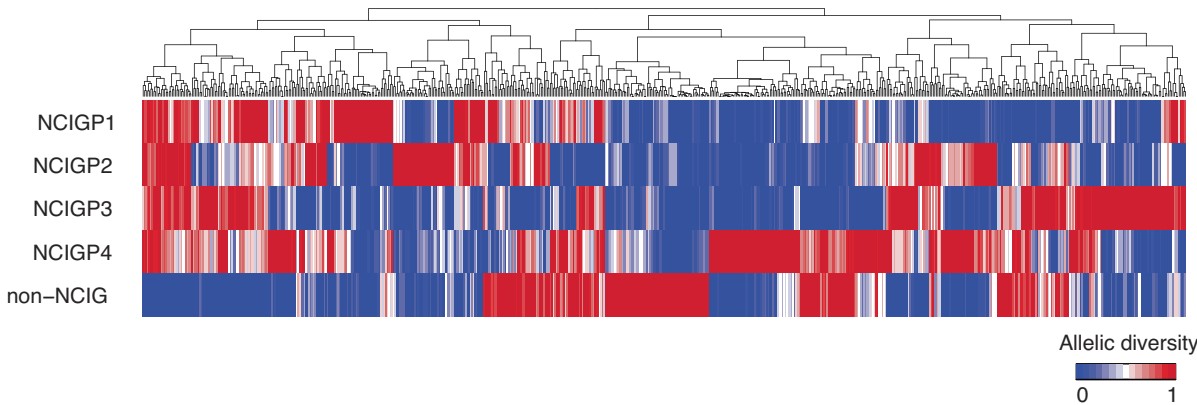

**Extended Data Fig. 10 | Analysis of STR expansion characteristics and community-specific variability.** (a) Dot plots show the number of repeat units versus the relative size increase of STR expansions of different period sizes. The horizontal dashed line indicates the minimum relative size increase (0.5) and the vertical dashed line indicates the minimum number of repeat units (10) required for an STR expansion to be further genotyped across all individuals in the cohort. (b) Matrix shows the normalised standard deviation (range 0:1) of allele sizes within each community for all STR sites with expansions in one or more individuals, in which allelic composition between the groups was significantly different. Hierarchical clustering was performed to group the STR sites based on the different patterns of variability between communities.

# Reporting Summary

## Statistics

For all statistical analyses, confirm that the following items are present in the figure legend, table legend, main text, or Methods section.

| n/a | Confirmed | |
|---|---|---|
| ☐ | ☒ | The exact sample size (*n*) for each experimental group/condition, given as a discrete number and unit of measurement |
| ☐ | ☒ | A statement on whether measurements were taken from distinct samples or whether the same sample was measured repeatedly |
| ☐ | ☒ | The statistical test(s) used AND whether they are one- or two-sided *Only common tests should be described solely by name; describe more complex techniques in the Methods section.* |
| ☐ | ☒ | A description of all covariates tested |
| ☐ | ☒ | A description of any assumptions or corrections, such as tests of normality and adjustment for multiple comparisons |
| ☐ | ☒ | A full description of the statistical parameters including central tendency (e.g. means) or other basic estimates (e.g. regression coefficient) AND variation (e.g. standard deviation) or associated estimates of uncertainty (e.g. confidence intervals) |
| ☐ | ☒ | For null hypothesis testing, the test statistic (e.g. *F*, *t*, *r*) with confidence intervals, effect sizes, degrees of freedom and *P* value noted *Give P values as exact values whenever suitable.* |
| ☒ | ☐ | For Bayesian analysis, information on the choice of priors and Markov chain Monte Carlo settings |
| ☒ | ☐ | For hierarchical and complex designs, identification of the appropriate level for tests and full reporting of outcomes |
| ☒ | ☐ | Estimates of effect sizes (e.g. Cohen's *d*, Pearson's *r*), indicating how they were calculated |

*Our web collection on statistics for biologists contains articles on many of the points above.*

## Software and code

Policy information about availability of computer code

| | |
|---|---|
| Data collection | Each library was loaded onto a PromethION flow cell (R9.4.1 for SQK-LSK110 libraries, R10.4.1 for SQK-LSK114 libraries) and sequenced on an ONT PromethION P48 device. Raw ONT sequencing data was converted from FAST5 to the more compact BLOW5 format in real-time on the PromethION during each sequencing run using slow5tools (v0.3.0). |
| Data analysis | Data was base-called using Guppy (v6.0.1) with the high-accuracy model and reads with mean quality < 7 were excluded from further analysis. The short read data was mapped using bwa-mem (v2.2.1) and the long read data was mapped using minimap2 (v2.22). Detection of large indels (20-49bp) and SVs (≥ 50bp) on short read mapped libraries was performed using smoove (v0.2.6) and on long read mapped libraries using CuteSV (v1.0.13). Individual callsets were then merged into a unified joint-call catalogue using Jasmine (v1.1.4). Indels and SVs were classified according to repeat type using custom analysis methods based on Tandem Repeat Finder (trf409.linux64) and RepeatMasker (4.1.2-p1). To assess the novelty of our SV catalogue, we compared SVs to: (i) the gnomAD (v2.1) SV database (ncbi.nlm.nih.gov/sites/dbvarapp/studies/nstd166/) and; (ii) an SV callset from population-scale ONT sequencing of Icelanders published recently by deCODE genetics (github.com/DecodeGenetics/LRS_SV_sets); (iii) and the Human Genome Structural Variation Consortium (HGSVC freeze 4; http://ftp.1000genomes.ebi.ac.uk/vol1/ftp/data_collections/HGSVC2). For individual-level, diploid genotyping of STR alleles, we also used custom scripts based on clair3 (v0.1-r12) and sniffles2 (v2.0.2) to detect variants and bcftools consensus (v1.12) to create haplotype-specific sequences. We used Centrifuge (v1.0.4) to identify and classify all non-human reads. We detected large CNVs (> 50 kb) in individual libraries using CNVpytor (version 1.3.1). The following additional tools were used during analysis: bedtools (2.28.0) UCSC LiftOver (kentUtils v302.1 ), R packages vegan (2.6.4), ape (5.7.1), hierfstat (0.5.11), stats (4.0.0). All data manipulation and visualisation, as well as plotting was performed in R (v4.0.0). All original code has been deposited at Zenodo and is publicly available as of the date of publication. |

For manuscripts utilizing custom algorithms or software that are central to the research but not yet described in published literature, software must be made available to editors and reviewers. We strongly encourage code deposition in a community repository (e.g. GitHub). See the Nature Portfolio guidelines for submitting code & software for further information.

# Data

Policy information about availability of data

All manuscripts must include a data availability statement. This statement should provide the following information, where applicable:

- Accession codes, unique identifiers, or web links for publicly available datasets
- A description of any restrictions on data availability
- For clinical datasets or third party data, please ensure that the statement adheres to our policy

The following publicly accessible datasets were used in this study:
(i) the gnomAD (v2.1) SV database: http://ncbi.nlm.nih.gov/sites/dbvarapp/studies/nstd166/
(ii) deCODE genetics SV callset: http://github.com/DecodeGenetics/LRS_SV_sets
(iii) HGSVC (freeze 4): http://ftp.1000genomes.ebi.ac.uk/vol1/ftp/data_collections/HGSVC2
The following reference genomes were used:
T2T-chm13 (v2.0): https://github.com/marbl/CHM13
Hg38 (GRCh38.p13): https://www.ncbi.nlm.nih.gov/datasets/genome/GCF_000001405.39/
All raw sequencing data, processed output files and associated metadata are permanently stored on Australia's National Computational Infrastructure (NCI) under the control of the Collection Access and Research Advisory Committee (CARAC) appointed and overseen by the National Centre for Indigenous Genomics (NCIG) Indigenous-majority governance board. Requests for access by external researchers will be considered by CARAC and governed by the NCIG Board. Data access requests from external researchers may be granted when the board is satisfied that core principles of Indigenous engagement are observed within the proposed research. At the heart of this is the requirement that the proposed research will be of benefit to Australian Indigenous peoples and is identified as important by the communities whose data is involved. Further information can be found within the NCIG governance framework:
https://ncig.anu.edu.au/files/NCIG-Governance-Framework.pdf
Data access requests should be directed to: jcsmr.ncig@anu.edu.au

# Human research participants

Policy information about studies involving human research participants and Sex and Gender in Research.

| Reporting on sex and gender | We collected sex information and the alignment of each individual library to either hg38 or T2T-chm13 was made in a sex-specific manner with an XY reference for genotypically male individuals and an XO reference for genotypically female individuals. Due to privacy concerns sex information of individuals was not made available, but sex information is also not required for the interpretation of any results presented in our manuscript. Other population characteristics covariate data, such as age, was not made available. |
|---|---|
| Population characteristics | We performed whole genome ONT sequencing on 121 individuals from four remote Aboriginal communities in northern Australia with whom the National Centre for Indigenous Genomics (NCIG) has developed partnerships: Tiwi Islands (n=41; Wurrumiyanga, Pirlangimpi and Millikapiti communities; NCIG-P1), Galiwin'ku (n=32; NCIG-P2), Titjikala (n=9; NCIG-P3) and Yarrabah (n=39; NCIG-P4). We also sequenced 18 non-Indigenous Australian individuals of European ancestry for comparison, and two reference individuals of European ancestry from the Genome in a Bottle project for control purposes (HG001, HG002). Across the cohort there were 79 genotypically female individuals and 62 genotypically male individuals. Other population characteristics covariate data, such as age, was not made available. |
| Recruitment | Appropriate permissions are sought from local governing bodies and community-led organizations to visit communities for discussing the Collection. The format, timing and place of discussions are determined by community members to ensure that their cultural perspectives and values are preserved and respected during conversations about personal or family samples held in the collection. Each participant provided informed consent (or assent for deceased kin) according to their individual legal rights and cultural perspectives to be a donor for the Collection. NCIG implemented a consent whereby the material and data can be reused for biomedical research and clinical applications. The researchers are not aware and did not control for potential self-selection or other biases during recruitment. |
| Ethics oversight | We are indebted to the individuals and their communities who participated in this research and to the National Centre for Indigenous Genomics (NCIG) Indigenous-majority Governance Board who helped guide this work in a culturally appropriate manner. The research was conducted in accordance with core principles of Indigenous community engagement, leadership and data sovereignty, as set out in the NCIG governance framework, approved under the Australian federal legislation: https://ncig.anu.edu.au/files/NCIG-Governance-Framework.pdf. Saliva and/or blood samples were collected from consenting individuals among four NCIG-partnered communities: Tiwi Islands (comprising the Wurrumiyanga, Pirlangimpi and Millikapiti communities), Galiwin'ku, Titjikala and Yarrabah, between 2015 and 2019. This study was approved by the Australian National University Human Research Ethics Committee (Ethics protocol number 2015/065). Non-Indigenous comparison data, generated from unrelated Australian individuals of European ancestry, was drawn from two existing biomedical research cohorts: (i) the Tasmanian Ophthalmic BioBank (Ethics protocol number 2020/ETH02479); and (ii) the Australian and New Zealand Registry of Advanced Glaucoma (Southern Adelaide Clinical Human Research Ethics Committee approval 305-08). |

Note that full information on the approval of the study protocol must also be provided in the manuscript.

# Field-specific reporting

Please select the one below that is the best fit for your research. If you are not sure, read the appropriate sections before making your selection.

☒ Life sciences          ☐ Behavioural & social sciences          ☐ Ecological, evolutionary & environmental sciences

For a reference copy of the document with all sections, see nature.com/documents/nr-reporting-summary-flat.pdf

# Life sciences study design

All studies must disclose on these points even when the disclosure is negative.

| | |
|---|---|
| Sample size | The samples sizes used in the study were not predetermined but the goal was to sequence as many individuals as possible to saturate discovery of structural variation in each of the indigenous communities. The number of participants included from each community group ranged from 9-41. Importantly, this encompassed and estimated 1-5% of all individuals within a given community. These large sample size ensure reliable representation of genetic diversity within each community that is necessary for population-scale analysis. |
| Data exclusions | No data were excluded from the analyses. |
| Replication | We sequenced 9-41 individuals from each community to ensure that our findings were representative of the variation present in those communities. Additionally, samples were split into 2 or more aliquots of 1mL each depending on the quantity of material available and one of the aliquots with 1mL sample was used for the DNA extraction and remaining aliquots were stored at -20 degrees Celcius or -80 degrees Celcius for long term storage. The stored aliquots can be accessed in the future to confirm any results if necessary. Additionally the indigenous samples have been previously & independently sequenced with short reads and that data can be used for crosschecks to resolve any potential sample swaps. Experimental replication has so far not been performed. |
| Randomization | This is not relevant as we did not allocate participants into experimental groups. |
| Blinding | Also not relevant because there was no group allocation of samples. |

# Reporting for specific materials, systems and methods

We require information from authors about some types of materials, experimental systems and methods used in many studies. Here, indicate whether each material, system or method listed is relevant to your study. If you are not sure if a list item applies to your research, read the appropriate section before selecting a response.

## Materials & experimental systems

| n/a | Involved in the study |
|---|---|
| ☒ ☐ | Antibodies |
| ☒ ☐ | Eukaryotic cell lines |
| ☒ ☐ | Palaeontology and archaeology |
| ☒ ☐ | Animals and other organisms |
| ☒ ☐ | Clinical data |
| ☒ ☐ | Dual use research of concern |

## Methods

| n/a | Involved in the study |
|---|---|
| ☒ ☐ | ChIP-seq |
| ☒ ☐ | Flow cytometry |
| ☒ ☐ | MRI-based neuroimaging |

