## [Peer Review File · Nature]

Manuscript Title: The landscape of genomic structural variation in Indigenous Australians

Reviewer Comments & Author Rebuttals

Reviewer Reports on the Initial Version:

Referees' comments:

Referee #1/2 (Remarks to the Author):

The authors present an overview of structural variation in Aboriginal Australian genomes. This is the first such study on Aboriginal Australians and naturally discovers many novel variants. I would have liked to have seen a clearer explanation of the significance (biological, population genetics, or medical) of the findings, which currently read a bit like a listing to me. I do not discern a clear narrative or conclusion.

I have some concerns that the background statements about the Aboriginal Australians need to be tightened to be more precise. For instance, the first sentence of the Introduction reads, "The Australian continent has the longest history of continuous human occupation outside Africa." How can this be true? Presumably the humans in Australia passed through another non-African continent, namely Asia, prior to reaching Australia. Did that continent (Asia) have discontinuous human occupation? (That is, was it uninhabited for large periods of time?) No. The fourth sentence of the introduction then asserts, also without any citation, that "Indigenous communities practice the world's oldest surviving cultures." What is meant here? Again as written this is a non-obvious, and I would say contentious assertion to make to an international audience. I would also reconsider the phrase "European invasion," as I do not think the authors are in fact suggesting that the continent of Europe invaded Australia. Perhaps be more precise throughout this prominent first paragraph.

The third paragraph of the Introduction refers to "exemplary frameworks" developed by NCIG. Is this an editorial opinion, or a fact? If the latter, it should be cited, if the former it seems out of place.

The last paragraph of the introduction refers to this study as "the first to do so on a minority and/or Indigenous cohort." What is meant by a "minority" population in this context? Vague nomenclature appears again on line 172, which refers to "clear genetic distinctions between Indigenous and non-Indigenous individuals." Almost certainly a similar distinction would be seen between Indigenous Australian and Indigenous American individuals, so it is not Indigenous vs. non-Indigenous that is the relevant distinction here, and I think this point is important, because this sentence could currently be copied and quoted without context in a damaging way. Perhaps one should say precisely what the groups are? Indigenous Australian vs. non-Indigenous Australian?

Why was Bray-Curtis dissimilarity chosen? This should be discussed.

If the main contribution of this paper, is the generation of data from Indigenous Australian individuals, then I'd like to see more clarity on whether this published data will be available for researchers. The current data availability statement that "All reasonable requests for access will be considered by CARAC, in accordance with principles of Indigenous data sovereignty" is quite unclear.

Referee #3 (Remarks to the Author):

1) Summary of the key results. Please summarise what you consider to be the outstanding features of the work.

The work describes the generation and analyses of long(er) DNA sequences using a very recent reference generated using long read sequencing in four sub populations of Indigenous Australians. The paper focus specifically on 'structural variation' - that is typically less well revealed by short(er) fragment sequencing technologies, and even less well by array-based genotyping methods. Using a recently assembled genome reference the authors identified structural variants identification and calling, focused analyses on SVs (≥ 50 bp, 159,912 thousand unique SVs) and large indels (136,797 of 20-49bp in size). The authors describe analyses of these data in terms of the following areas:

- i) Landscape of structural variation within and among four Indigenous Australian populations
- ii) Functional significance of a sub-class of genomewide SVs (short tandem repeats, STRs) in disease associated genes that revealed novel genetic variation and allelic diversity within expressed genes, as well as specific tests for evidence of putative purifying selection.
- iii) Application of the information generated in an Aboriginal health context.

The study describes relatively high levels of structural variation that appears to be higher levels of sub-population specificity indicating the need for multiple rather than single reference genomes for indigenous Australian populations.

I consider the following to be outstanding features of the paper:

- The extent of genetic diversity at the structural variant level and its distribution within and between populations, and implications for genomic studies in health related conditions in indigenous Australians, and
- In particular the potential contribution of structural variation to heritable conditions, including prioritising this via the detection of STRs associated with a specific disease in an individual which demonstrates the role of specific genomic technologies (in this case, long read sequencing) to delivering diagnoses and therefore potential interventions in populations historically underserved by genomic sciences and negatively impacted by colonisation. This is a novel combination, and more aligned with various indigenous ethical frameworks which prioritise benefits to communities above basic/fundamental science.

2) Validity: Does the manuscript have flaws which should prohibit its publication? If so, please provide details.

I do not consider the manuscript contains major flaws that prohibits its publication (subject to caveats regarding areas listed below where I cannot provide expert opinion or knowledgeable

insights), although one area that should be included in my view are measures of within and between population variation. Standard methods for quantifying inter and inter-population variation (FIS, FST etc – or GST for multi allelic loci) are not reported thus comparisons with other studies involving indigenous Australians as well as other Pacific populations are limited. There are also some minor issues that do need to be addressed (below).

3) Originality and significance: if not novel, please include reference. If the conclusions are not original, please provide relevant references. On a more subjective note, do you feel that the results presented are of immediate interest to many people in your own discipline, and/or to people from several disciplines?

a) Originality:

i) The key original contribution in my view is the combination of functional significance of and population genetics of structural variation, along with the biological significance and the path to delivery of benefit to study participants (and by implication, communities).

[Less original is the population genetics significance: previous studies have more deeply trawled the relative contributions of within vs among-population variation (e.g. Malaspina, A.-S. et al. A genomic history of Aboriginal Australia. *Nature* 538, 207–214 (2016)). Results are perhaps not so surprising in that they replicate other types of DNA-based variation. The authors may wish to comment on what is it about structural variation that differs from other categories of variation (e.g., SNPs).]

ii) The nature and role of structural variation in these populations relative to both each other and other populations mentioned. To date few long read-based population genomics studies have been reported, and this study provides some important differences to those studies already reported on, which have consequences for reference genome construction and utilisation. The research goes on to show 'Indeed, the incidental identification of a pathogenic MJD expansion in one individual highlights the strength of our approach in this domain.'

a) Significance: are the results only of immediate interest to own discipline or people from several disciplines?

i) I consider these results to be of interest to people in the following sub-disciplines:

(1) Human genomic structures, particularly structures of unique populations/underrepresented groups in international databases;

(2) genomics/health interface;

(3) population genetics of human populations;

(4) heritable disease aetiology, particularly the impact of structural variation; and

(5) exemplars of direct health relevance and benefits to indigenous populations from genomic technologies. [I also note there are may be other sub disciplines that would be interested in this work, e.g., structural biochemists, looking at novel protein structure variants in disease related genes].

ii) Broader appeal could be achieved by undertaking additional analyses and providing additional information re: evolutionary histories: the population group itself inhabited Australia over 50,000 years ago without much admixture from other groups subsequently. The Denisovan contribution differs too other groups in Austronesia. The within population genetic differentiation, whilst previously known, has not been explored at a genome structure-level for this category of structural variation. Investigating the contribution of Denisovan genomes and comparing structural variation in

the Denisovan and modern humans (in particular to disease-associated loci) would broaden appeal of this paper to those interested in evolution of human populations, in particular the contribution of SVs.

[However, from an indigenous ethics point of view, the health implications of structural variation and the derived information and demonstrable pathways to applications of that information is of paramount importance, therefore these additions are of more scientific interest only, and the authors are correct to prioritise to health implications.]

b) Additional information:

i) Methods used to engage and successfully recruit participants, and classification of that component of study design would be useful. In this regard, my understanding is that samples were not collected for this specific study, but rather, for previous studies involving these communities, and that such 'legacy samples' had limited applicability to ethically addressing equity issues in health. The re-purposing of samples these samples in manner more likely to achieve community benefit

2) Data & methodology: validity of approach, quality of data, quality of presentation. Please comment on the validity of the approach, quality of the data and quality of presentation. Please note that we expect our reviewers to review all data, including any extended data and supplementary information. Is the reporting of data and methodology sufficiently detailed and transparent to enable reproducing the results?

a) Validity of approach:

i) Data acquisition: see comment above regarding recruitment

ii) DNA extraction:

(1) Saliva samples often contain non-human DNA. How was this detected and masked? Were the variant calling and quality filtering procedures robust enough to eliminate non-human eucaryotic DNA that could give rise to false variants? How was this checked?

iii) Reference alignment and SV mapping:

(1) against the most recent reference (T2T) was appropriate.

(2) [I was not able to complete my review owing to health reasons so could not provide a complete evaluation of analytical methods used. HOWEVER no methods seemed inappropriate. Perhaps selection analyses using haplotype methods (iHS etc) could have been undertaken to provide a more complete picture of selection.]

b) Quality of presentation is very high. I do have specific edits to figures or figure legends:

i) Figure 4A – add the percent variance explained by the two {Principal components [CHECK where the PCs are listed]}

ii) Figure 4b – reorder the bars to reflect the individual, pairwise and community overlaps, in that order. The existing order was confusing and seemed to be designed on the basis of visual appeal

3) Appropriate use of statistics and treatment of uncertainties. All error bars should be defined in the corresponding figure legends; please comment if that's not the case. Please include in your report a specific comment on the appropriateness of any statistical tests, and the accuracy of the description of any error bars and probability values.

a) Methods for reference alignment etc uses statistical inference. These are not my specific area of expertise therefore I defer to other reviewers.

4) Conclusions: robustness, validity, reliability. Do you find that the conclusions and data interpretation are robust, valid and reliable?

a) Yes

5) Suggested improvements: experiments, data for possible revision. Please list additional experiments or data that could help strengthening the work in a revision.

a) See (c) above

b) Also, there is no reporting of the use of standard statistical methods for assessing population differentiation (multiallelic equivalents of FST, FIS etc) and therefore the combination of population/variant type is difficult to compare with other population studies indigenous Australians as well as indigenous Pacific populations. For example, how might these compare with genomewide estimates from other Pacific populations (e.g., Cadzow et al who provided estimates in other indigenous Pacific populations such as Māori, Samoan etc based on SNP array data?

6) References: appropriate credit to previous work? Does this manuscript reference previous literature appropriately? If not, what references should be included or excluded?

a) Previous work on the population structure in indigenous Australians is referenced

b) Also referencing regarding previous examples of STR on disease (although Huntington's – a classic example – is not mentioned...)

c) Perhaps more on indigenous ethical frameworks for research such as but not limited to medical genomics

7) Clarity and context: lucidity of abstract/summary, appropriateness of abstract, introduction and conclusions. Is the abstract clear, accessible? Are abstract, introduction and conclusions appropriate?

a) Lucidity of abstract/summary: yes – sufficiently clear in my opinion

b) Is the abstract clear, accessible? And appropriateness of abstract, introduction and conclusions: yes – sufficiently clear in my opinion

c) Some minor edits suggested:

i) Frequent use of 'Data was' instead of 'Data were'

8) Inflammatory material: Does the manuscript contain any language that is inappropriate or potentially libelous?

No.

10) Please indicate any particular part of the manuscript, data, or analyses that you feel is outside the scope of your expertise, or that you were unable to assess fully.

a) I consider the following areas are outside of my specific expertise, therefore rely on other reviewers to provide more informed insight:

i) Specifics workflows regarding the genome assemblies from long read data.

ii) Nuances of the LOEUF method.

b) Also, I feel it important to point out that I reviewed this paper whilst impacted by Covid-19, in particular episodic bouts of fatigue following infection in late February. While I do not feel this

reduced the quality of my review, it did both delay and truncate it. I apologise to the authors and the editors for the inconvenience.

Referee #4 (Remarks to the Author):

Reis and coauthors present the sequencing and analysis of 141 Indigenous Australian individuals using long-read sequencing. This is a valuable resource reflecting in-depth analysis of a population that is underrepresented in existing diversity studies (1000-genomes). The analysis described in this manuscript include: description of counts of variants (landscape), population frequency, functional impact (CDS and LOEUF analysis), and STR analysis. An example of a potentially pathogenic STR expansion in ATXN3.

While this reflects a landmark study on genetic diversity of indigenous populations, there are two areas of critique, methodological and the scope of the analysis that was performed.

Methodology:

1. The abstract states that 73% of variants are not previously annotated, and the results state 965 NCIG only. This high number is likely an artifact of how uniqueness of a variant is calculated. As the authors state, many of the SV are tandem repeats. It is difficult to assess SV novelty should be assessed with respect to both gnomAD as well as variants from the Human Genome Structural Variation Consortium (HGSVC). While the deCODE dataset is large, it reflects the Icelandic population and the relatively low population diversity of that cohort. Optionally, the variants can be compared to those derived from the Human Pangenome Reference Consortium (HPRC), however this study has only been released in preprint: <https://www.biorxiv.org/content/10.1101/2022.07.09.499321v1>. The deCODE callset is fairly comprehensive, however the HGSVC calls are closer to base-level accuracy.

Furthermore, there is an estimate that 2,884 SVs per individual are NCIG-only. This and the corresponding statistic is influenced by the cohorts this callset is compared against.

2. The 80% reciprocal similarity is too high for tandem repeats, or reflects an arbitrary value depending on how different alleles are counted. This may overstate the amount of novel variation in these genomes. One potential metric is to compare the number of all novel SV to only Alu MEI. The Alu insertions are well defined (typically), and can help gauge the extent that novel variation is influenced by this high reciprocal overlap. Furthermore, the number of distinct SV (not near other SV) should be distinguished from SV at the same locus. One approach is to identify tandem repeat loci in CHM13 and apply a special merging of variants per locus, rather than using the reciprocal overlap.

The number of unique SV should be counted as done in the current methods, but also only on the SV that are not proximal to others (within a few kb, or not near the same annotated tandem repeat on CHM13). The authors should clearly state what is a new SV: does this include different alleles on the same locus?

The read-based calling using cute-SV is subject to noise in calling SV near tandem repeats. An alternative to this is using de novo assembly to first assemble genomes, then call variants from assemblies. This is the approach taken by both the HGSVC and HPRC.

3. Some analysis of the variant calling accuracy should be performed for the data used. Since ONT is continually evolving it is important to provide a benchmark for the study. The overall results for variant calls per individual are within expectation. Still, the results should begin with 2-3 sentences for validation of the sequencing and variant calling strategy on the HG002 reference. The diploid reference from <https://www.nature.com/articles/s41586-022-05325-5> can be used as ground truth, and the number of variants called from the reference should be compared to the alignment+cutesv pipeline at 15-50X coverage (by 5), in order to study the relationship between coverage and variant calls a the range of coverages generated on this cohort. Also, a supplementary figure for variants called per genome by coverage should be included.

While long reads enable improved SV discovery over short reads, there are some analyses that are missing from this study that could improve its impact:

1. Copy number analysis. Cute-SV does not detect copy number variants (and is particularly bad at discovering rearrangements in general). It is straight forward to identify copy number variable genes through excess mapped read depth.
2. Fixation index (FST) or VST can be calculated on this cohort relative to HPRC+HGSVC data on non-tandem repeat loci when compared to the decode calls.

Less necessary:

3. A comparison of SNP and SV calls in this population. The fraction of population specific SNVs can be used to contrast the unique SVs, and can help give context for how accurate the estimate of population specific SVs.
4. Assemblies of HLA regions could give insight to innate/adaptive immunity in this population.

REVIEWERS #1/2 (Expertise: human evolutionary genetics)

1.1. The authors present an overview of structural variation in Aboriginal Australian genomes. This is the first such study on Aboriginal Australians and naturally discovers many novel variants. I would have liked to have seen a clearer explanation of the significance (biological, population genetics, or medical) of the findings, which currently read a bit like a listing to me. I do not discern a clear narrative or conclusion.

We thank the reviewers for their careful appraisal of our manuscript. Below we outline our responses to all comments and concerns they have raised. We hope this has improved the clarity of the narrative.

1.2. I have some concerns that the background statements about the Aboriginal Australians need to be tightened to be more precise. For instance, the first sentence of the Introduction reads, "The Australian continent has the longest history of continuous human occupation outside Africa." How can this be true? Presumably the humans in Australia passed through another non-African continent, namely Asia, prior to reaching Australia. Did that continent (Asia) have discontinuous human occupation? (That is, was it uninhabited for large periods of time?).

Archaeological and genomic evidence indicates that modern Aboriginal Australians are direct descendants of the ancestral human population that first colonized Australia, around ~50 thousand years ago¹⁻³. While modern humans may have settled other non-African locations, and necessarily passed through Asia to reach Australia, these early populations were replaced by more recent populations, making their cultural and genetic legacy less continuous than Indigenous Australians⁴.

While there is evidence to support our statement on continuous human settlement in Australia, we accept that it is somewhat oversimplified and open to misinterpretation. To mitigate this, and because continuous human settlement is not a central subject of the paper, we have decided to remove this sentence.

Actions: The sentence in question has been removed to avoid potential misinterpretation.

1.3. The fourth sentence of the introduction then asserts, also without any citation, that "Indigenous communities practice the world's oldest surviving cultures." What is meant here? Again as written this is a non-obvious, and I would say contentious assertion to make to an international audience.

Again, we accept that this short statement is an over-simplification. Evidence suggests Indigenous languages, knowledge, customs and beliefs have been passed down through oral history since the earliest settlement of the Australian continent^{5,6}. While this is remarkable, there are other peoples (e.g. the San in southern Africa) who might lay claim to cultural history of similar duration, and there is really no need to try and pick a winner here. Therefore, to avoid possible contention, we have modified this sentence as follows:

Australian Indigenous communities practice cultures that are among the world's oldest continuous surviving cultures.

Actions: The sentence in question has been amended for improved clarity/precision.

1.4. I would also reconsider the phrase "European invasion," as I do not think the authors are in fact suggesting that the continent of Europe invaded Australia. Perhaps be more precise throughout this prominent first paragraph.

We stand by the use of the term 'invasion', rather than common alternatives like 'occupation' or 'settlement', which imply that the Australian continent was uninhabited prior to the arrival of Europeans (it was not). However, we have substituted 'European invasion' for the more precise phrase 'invasion by people from Europe'.

We invite the reviewer to consider the introductory paragraph in full, now revised to address points **1.2 - 1.4**:

Australia is home to hundreds of Aboriginal nations or clans who inhabited all geographical regions throughout the continent, prospering in their diverse environments for at least fifty thousand years¹⁻⁵. Over 250 distinct languages were spoken at the time of invasion by people from Europe and ~150 of these survive today⁶. Australian Indigenous communities practice cultures that are among the world's oldest continuous surviving cultures. These are highly varied, but commonly emphasise the importance of kinship, ancestry, and relationships to the landscape and environment⁵.

Actions: The opening paragraph has been amended for improved clarity/precision.

1.5. The third paragraph of the Introduction refers to "exemplary frameworks" developed by NCIG. Is this an editorial opinion, or a fact? If the latter, it should be cited, if the former it seems out of place.

We accept that the term 'exemplary' is opinion rather than fact and is unnecessary here. The sentence now reads:

The NCIG has developed frameworks for Indigenous genomics that prioritise community leadership, participation and data sovereignty.

Actions: The term 'exemplary' has been removed from the paragraph in question.

1.6. The last paragraph of the introduction refers to this study as "the first to do so on a minority and/or Indigenous cohort." What is meant by a "minority" population in this context?

The handful of other population-scale long-read sequencing studies published so far, including most prominently the Icelandic study from deCODE, have focused on comparatively homogeneous, well-studied populations and primarily sampled the majority ancestry group in the country where the study was conducted⁷⁻⁹. Ours is unique in targeting diverse communities (specifically Indigenous communities) who are typically under-represented in genomics research. However, we accept that the term 'minority population' is somewhat vague and unnecessary here and we have therefore removed it. The sentence now reads:

Our study is among the first to deploy long-read sequencing at population-scale and the first to do so in a diverse Indigenous cohort¹⁹.

Actions: The term 'minority' has been removed from the paragraph in question.

1.7. Vague nomenclature appears again on line 172, which refers to "clear genetic distinctions between Indigenous and non-Indigenous individuals." Almost certainly a similar distinction would be seen between Indigenous Australian and Indigenous American individuals, so it is not Indigenous vs. non-Indigenous that is the relevant distinction here, and I think this point is important, because this sentence could currently be copied and quoted without context in a damaging way. Perhaps one should say precisely what the groups are? Indigenous Australian vs. non-Indigenous Australian?

We agree that the wording on this sentence was imprecise and could be quoted without appropriate context. We have revised the sentence to:

The clear genetic distinctions between Indigenous Australian and non-Indigenous Australian individuals was further reiterated by Principal coordinate (PCoA) and Fixation Index (F_{ST}) analysis of structural variation (Fig4a; Extended Data Fig6a).

Actions: The sentence in question has been amended for improved clarity/precision.

1.8. Why was Bray-Curtis dissimilarity chosen? This should be discussed.

Bray-Curtis similarity is a widely used statistical measure that can be used to assess the genetic distance between groups of individuals. This measure is based on the relative abundance of traits, in this case structural variant (SV) alleles, across diverse individuals. The Bray-Curtis method has several advantages in this context:

- it measures the relative abundance of shared SVs between individuals or populations, rather than simply whether an SV is present or absent, thereby providing greater resolution to distinguish closely related individuals;
- it can handle missing data, which makes it more robust to variables like different sequencing coverage, read-length or data quality that may impact SV detection in different individuals/samples;
- the method has been used previously with various types of genetic data including SNPs, microsatellites, and (importantly) SVs. For example - see Sudmant *et al* (2015) PMID:26432246

Actions: We have added a paragraph to the **Methods** section explaining our use of Bray-Curtis similarity.

1.9. If the main contribution of this paper, is the generation of data from Indigenous Australian individuals, then I'd like to see more clarity on whether this published data will be available for researchers. The current data availability statement that "All reasonable requests for access will be considered by CARAC, in accordance with principles of Indigenous data sovereignty" is quite unclear.

We have added further information to the data availability statement regarding NCIG data sharing criteria. The full statement reads:

All raw sequencing data, processed output files and associated metadata are permanently stored on Australia's National Computational Infrastructure (NCI) under the control of the Collection Access and Research Advisory Committee (CARAC) appointed and overseen by the National Centre for Indigenous Genomics (NCIG) Indigenous-majority governance board. Requests for access by external researchers will be considered by CARAC and governed by the NCIG Board. Data access requests from external researchers may be granted when the board is satisfied that core principles of Indigenous engagement are observed within the proposed research. At the heart of this is the requirement that the proposed research will be of benefit to Australian Indigenous peoples and is identified as important by the communities whose data is involved. Further information can be found within the NCIG governance framework:

<https://ncig.anu.edu.au/files/NCIG-Governance-Framework.pdf>

Data access requests should be directed to: jcsmr.ncig@anu.edu.au

Actions: 'Data Availability' statement updated to better describe data sharing criteria.

REVIEWER #3 (Expertise: Indigenous genomics)

We thank the reviewer for their careful appraisal of our manuscript. Below we outline our responses to all comments and concerns they have raised.

2.1. Summary of the key results. Please summarise what you consider to be the outstanding features of the work.

The work describes the generation and analyses of long(er) DNA sequences using a very recent reference generated using long read sequencing in four sub populations of Indigenous Australians. The paper focus specifically on 'structural variation' - that is typically less well revealed by short(er) fragment sequencing technologies, and even less well by array-based genotyping methods. Using a recently assembled genome reference the authors identified structural variants identification and calling, focused analyses on SVs (≥ 50 bp, 159,912 thousand unique SVs) and large indels (136,797 of 20-49bp in size). The authors describe analyses of these data in terms of the following areas:

- i) Landscape of structural variation within and among four Indigenous Australian populations

ii) Functional significance of a sub-class of genomewide SVs (short tandem repeats, STRs) in disease associated genes that revealed novel genetic variation and allelic diversity within expressed genes, as well as specific tests for evidence of putative purifying selection.

iii) Application of the information generated in an Aboriginal health context.

The study describes relatively high levels of structural variation that appears to be higher levels of sub-population specificity indicating the need for multiple rather than single reference genomes for indigenous Australian populations.

I consider the following to be outstanding features of the paper:

- The extent of genetic diversity at the structural variant level and it's distribution within and between populations, and implications for genomic studies in health related conditions in indigenous Australians, and

- In particular the potential contribution of structural variation to heritable conditions, including prioritising this via the detection of STRs associated with a specific disease in an individual which demonstrates the role of specific genomic technologies (in this case, long read sequencing) to delivering diagnoses and therefore potential interventions in populations historically underserved by genomic sciences and negatively impacted by colonisation. This is a novel combination, and more aligned with various indigenous ethical frameworks which prioritise benefits to communities above basic/fundamental science.

Thank you. This summary appropriately captures the content and significance of our study.

Actions: No action required.

2.2. Validity: Does the manuscript have flaws which should prohibit its publication? If so, please provide details.

I do not consider the manuscript contains major flaws that prohibits its publication (subject to caveats regarding areas listed below where I cannot provide expert opinion or knowledgeable insights), although one area that should be included in my view are measures of within and between population variation. Standard methods for quantifying inter and inter-population variation (FIS, FST etc – or GST for multi allelic loci) are not reported thus comparisons with other studies involving indigenous Australians as well as other Pacific populations are limited. There are also some minor issues that do need to be addressed (below).

We are happy that the reviewer could find no major flaws in the validity of our study. At their request, we have added an analysis of inter-community variation based on Fixation Index (F_{ST}) to the **Results** section (under the 'Diversity and Distribution' subheading). We used F_{ST} to measure the genetic distances between NCIG (and non-NCIG) communities, and pairwise F_{ST} comparisons are summarized in **Extended Data Fig6a** (shown below, right panel). This analysis further reiterates the clear genetic distinctions between Indigenous and non-Indigenous Australian individuals, and between different Indigenous communities, which were evident in the PCoA analysis presented in **Fig4a** (shown below, left panel). The relative pairwise distances between groups are broadly similar for F_{ST} and PCoA analyses:

Actions: New analysis of genetic distances between communities using Fixation index (F_{ST}) was performed; F_{ST} results are reported in **Extended Data Fig6a**.

2.3. Originality and significance: if not novel, please include reference. If the conclusions are not original, please provide relevant references. On a more subjective note, do you feel that the results presented are of immediate interest to many people in your own discipline, and/or to people from several disciplines?

a) Originality:

i) The key original contribution in my view is the combination of functional significance of and population genetics of structural variation, along with the biological significance and the path to delivery of benefit to study participants (and by implication, communities).

[Less original is the population genetics significance: previous studies have more deeply trawled the relative contributions of within vs among-population variation (e.g. Malaspina, A.-S. et al. A genomic history of Aboriginal Australia. Nature 538, 207–214 (2016)). Results are perhaps not so surprising in that they replicate other types of DNA-based variation.]

ii) The nature and role of structural variation in these populations relative to both each other and other populations mentioned. To date few long read-based population genomics studies have been reported, and this study provides some important differences to those studies already reported on, which have consequences for reference genome construction and utilisation. The research goes on to show ‘Indeed, the incidental identification of a pathogenic MJD expansion in one individual highlights the strength of our approach in this domain.’

We agree with this appraisal of our study’s originality.

Actions: No action required.

2.4. a) Significance: are the results only of immediate interest to own discipline or people from several disciplines?

i) I consider these results to be of interest to people in the following sub-disciplines:

(1) Human genomic structures, particularly structures of unique populations/underrepresented groups in international databases;

(2) genomics/health interface;

(3) population genetics of human populations;

(4) heritable disease aetiology, particularly the impact of structural variation; and

(5) exemplars of direct health relevance and benefits to indigenous populations from genomic technologies. [I also note there are may be other sub disciplines that would be interested in this work, e.g., structural biochemists, looking at novel protein structure variants in disease related genes].

ii) Broader appeal could be achieved by undertaking additional analyses and providing additional information re: evolutionary histories: the population group itself inhabited Australia over 50,000 years ago without much admixture from other groups subsequently. The Denisovan contribution differs to other groups in Austronesia. The within population genetic differentiation, whilst previously known, has not been explored at a genome structure-level for this category of structural variation. Investigating the contribution of Denisovan genomes and comparing structural variation in the Denisovan and modern humans (in particular to disease-associated loci) would broaden appeal of this paper to those interested in evolution of human populations, in particular the contribution of SVs.

[However, from an indigenous ethics point of view, the health implications of structural variation and the derived information and demonstrable pathways to applications of that information is of paramount importance, therefore these additions are of more scientific interest only, and the authors are correct to prioritise to health implications.]

We agree with this appraisal of our study's significance and broad appeal. We also agree that an analysis of Denisovan admixture using the long-read sequencing data that we have generated may lead to new insights. However, we have a **strict ethical obligation to only undertake research that is of direct benefit to Indigenous Australians, and addresses questions on which the partner communities have been consulted and provided approval**. While a Denisovan admixture analysis would be scientifically interesting, this was not proposed during community consultations for our study and has not been approved by our partners. Therefore, we prefer to focus on the health implications of genomic structural variation, which the reviewer correctly identifies as the top priority here.

Actions: Suggested analysis is out of scope of approved research.

2.5. Additional information:

i) Methods used to engage and successfully recruit participants, and classification of that component of study design would be useful. In this regard, my understanding is that samples were not collected for this specific study, but rather, for previous studies involving these communities, and that such 'legacy samples' had limited applicability to ethically addressing equity issues in health. The re-purposing of samples these samples in manner more likely to achieve community benefit.

Our study uses samples from the National Centre for Indigenous Genomics (NCIG) collection, which encompasses historic and contemporary biospecimens, records and genomic data under Indigenous governance and custodianship. The collection includes short-read whole-genome sequencing data on 159 Indigenous Australians generated previously, which are the subject of another manuscript currently under consideration at *Nature*. With the permission and ongoing oversight of the relevant communities and NCIG board, we gained access to suitable DNA samples obtained from saliva and blood for our research project on genomic structural variation.

Community consultation, consent and collection processes were not included in the **Methods** section of our paper because they were described elsewhere. We have added citations to two articles where the reader can find more detailed information about the NCIG collection (historical and contemporary) and NCIG's culturally-appropriate community consultation practices:

Lewis, D. Australian biobank repatriates hundreds of 'legacy' Indigenous blood samples. *Nature Publishing Group UK*

Hermes, A. et al. Beyond platitudes: a qualitative study of Australian Aboriginal people's perspectives on biobanking. *Intern. Med. J.* 51, 1426–1432 (2021).

Actions: Additional references added to **Introduction** to provide further information about NCIG collection and community engagement practices.

2.6. Data & methodology: validity of approach, quality of data, quality of presentation. Please comment on the validity of the approach, quality of the data and quality of presentation. Please note that we expect our reviewers to review all data, including any extended data and supplementary information. Is the reporting of data and methodology sufficiently detailed and transparent to enable reproducing the results?

a) Validity of approach:

i) Data acquisition: see comment above regarding recruitment

ii) DNA extraction:

(1) Saliva samples often contain non-human DNA. How was this detected and masked? Were the variant calling and quality filtering procedures robust enough to eliminate non-human eucaryotic DNA that could give rise to false variants? How was this checked?

The reviewer is correct that saliva DNA samples may contain non-human DNA; we found an average of $6.04\% \pm 5.57$ non-human reads in saliva samples in our cohort, compared to just $0.08\% \pm 0.06$ for blood samples. Such contamination can be a source of error in genomics studies using short-read sequencing. However, long-read sequencing data has higher sequence specificity, reducing the likelihood of erroneous alignment of non-human reads to the human genome. To confirm this, we have performed a new analysis in which we identified and classified non-human reads in all samples using *Centrifuge*¹⁰, then measured the rate of alignment for these reads to the *chm13* genome under our standard workflow. The fraction of non-human reads that were erroneously aligned to human chromosomes was negligible: mean $4.23\% (\pm 1.90)$, or just $0.2\% (\pm 0.14)$ of all reads in a given library. We refer the reviewer to the analysis of non-human reads presented below, which has been added to **Extended Data Fig1a,b**. Non-human read counts for each library are also now reported in **Supplementary Table 1**.

a Non-human reads

b Alignment of non-human reads

Actions: new analysis showing negligible mapping of non-human reads presented in **Extended Data Fig1a,b**; unmapped read counts per sample reported in **Supplementary Table 1**.

2.7. iii) Reference alignment and SV mapping:

(1) against the most recent reference (T2T) was appropriate.

(2) [I was not able to complete my review owing to health reasons so could not provide a complete evaluation of analytical methods used. HOWEVER no methods seemed inappropriate. Perhaps selection analyses using haplotype methods (iHS etc) could have been undertaken to provide a more complete picture of selection.]

b) Quality of presentation is very high. I do have specific edits to figures or figure legends:

i) Figure 4A – add the percent variance explained by the two {Principal components [CHECK where the PCs are listed]}

ii) Figure 4b – reorder the bars to reflect the individual, pairwise and community overlaps, in that order. The existing order was confusing and seemed to be designed on the basis of visual appeal

Thank you for these suggestions, we have made the suggested changes.

Actions: Percent variance added to axes and legend for Fig4a; bars reordered in Fig4b.

2.8. Appropriate use of statistics and treatment of uncertainties. All error bars should be defined in the corresponding figure legends; please comment if that's not the case. Please include in your report a specific comment on the appropriateness of any statistical tests, and the accuracy of the description of any error bars and probability values.

a) Methods for reference alignment etc uses statistical inference. These are not my specific area of expertise therefore I defer to other reviewers.

No reviewers have raised concerns in regard to the methods used for reference alignment.

Actions: no action required.

4) Conclusions: robustness, validity, reliability. Do you find that the conclusions and data interpretation are robust, valid and reliable?

a) Yes

5) Suggested improvements: experiments, data for possible revision. Please list additional experiments or data that could help strengthening the work in a revision.

a) See (c) above

b) Also, there is no reporting of the use of standard statistical methods for assessing population differentiation (multiallelic equivalents of F_{ST} , F_{IS} etc) and therefore the combination of population/variant type is difficult to compare with other population studies indigenous Australians as well as indigenous Pacific populations. For example, how might these compare with genomewide estimates from other Pacific populations (e.g., Cadzow et al who provided estimates in other indigenous Pacific populations such as Māori, Samoan etc based on SNP array data?

This is addressed through the addition of new analysis quantifying genetic distances using a Fixation index (F_{ST}) method, as described above (see **2.2**).

Actions: New analysis of genetic distances between communities using Fixation index (F_{ST}) was performed; F_{ST} results are reported in Extended Data Fig6a.

6) References: appropriate credit to previous work? Does this manuscript reference previous literature appropriately? If not, what references should be included or excluded?

a) Previous work on the population structure in indigenous Australians is referenced

b) Also referencing regarding previous examples of STR on disease (although Huntington's – a classic example – is not mentioned...)

c) Perhaps more on indigenous ethical frameworks for research such as but not limited to medical genomics

At the reviewer's suggestion, we have included additional references describing the NCIG's Indigenous ethical frameworks and community consultation practices:

Lewis, D. Australian biobank repatriates hundreds of 'legacy' Indigenous blood samples. *Nature Publishing Group UK*

Hermes, A. et al. Beyond platitudes: a qualitative study of Australian Aboriginal people's perspectives on biobanking. *Intern. Med. J.* 51, 1426–1432 (2021).

Actions: additional references on NCIG's Indigenous ethical frameworks and community engagement added to Introduction.

7) Clarity and context: lucidity of abstract/summary, appropriateness of abstract, introduction and conclusions. Is the abstract clear, accessible? Are abstract, introduction and conclusions appropriate?

a) Lucidity of abstract/summary: yes – sufficiently clear in my opinion

b) Is the abstract clear, accessible? And appropriateness of abstract, introduction and conclusions: yes – sufficiently clear in my opinion

c) Some minor edits suggested:

i) Frequent use of 'Data was' instead of 'Data were'

Thank you for identifying this error; this has been corrected throughout.

Actions: 'Data was' corrected to 'Data were' throughout the manuscript.

8) Inflammatory material: Does the manuscript contain any language that is inappropriate or potentially libelous?

No.

10) Please indicate any particular part of the manuscript, data, or analyses that you feel is outside the scope of your expertise, or that you were unable to assess fully.

a) I consider the following areas are outside of my specific expertise, therefore rely on other reviewers to provide more informed insight:

i) Specifics workflows regarding the genome assemblies from long read data.

ii) Nuances of the LOEUF method.

b) Also, I feel it important to point out that I reviewed this paper whilst impacted by Covid-19, in particular episodic bouts of fatigue following infection in late February. While I do not feel this reduced the quality of my review, it did both delay and truncate it. I apologise to the authors and the editors for the inconvenience.

We're sorry to hear that the reviewer was impacted by COVID-19. We appreciate their efforts to complete the review and wish them the best.

Actions: No action required.

REVIEWER #4 (Expertise: long-read sequencing, SV)

3.1. Reis and coauthors present the sequencing and analysis of 141 Indigenous Australian individuals using long-read sequencing. This is a valuable resource reflecting in-depth analysis of a population that is underrepresented in existing diversity studies (1000-genomes). The analysis described in this manuscript include: description of counts of variants (landscape), population frequency, functional impact (CDS and LOEUF analysis), and STR analysis. An example of a potentially pathogenic STR expansion in ATXN3.

While this reflects a landmark study on genetic diversity of indigenous populations, there are two areas of critique, methodological and the scope of the analysis that was performed.

We thank the reviewer for their careful appraisal of our manuscript. Below we outline our responses to all comments and concerns they have raised.

3.2. The abstract states that 73% of variants are not previously annotated, and the results state 965 NCIG only. This high number is likely an artifact of how uniqueness of a variant is calculated. As the authors state, many of the SV are tandem repeats. It is difficult to

SV novelty should be assessed with respect to both gnomAD as well as variants from the Human Genome Structural Variation Consortium (HGSVC). While the deCODE dataset is large, it reflects the Icelandic population and the relatively low population diversity of that cohort. Optionally, the variants can be compared to those derived from the Human Pangenome Reference Consortium (HPRC), however this study has only been released in preprint: <https://www.biorxiv.org/content/10.1101/2022.07.09.499321v1>. The deCODE callset is fairly comprehensive, however the HGSVC calls are closer to base-level accuracy.

As suggested, we have compared our SV callset to the latest SV annotation (Freeze 4) from the Human Genome Structural Variation Consortium (HGSVC), in addition to the gnomAD and deCODE annotations that we used originally. We have also refined the criteria used to identify whether an equivalent or similar SV was present in any of these annotations (see **3.4**, below). This updated analysis is presented in **Fig2e**.

Although HGSVC Freeze 4 encompasses long-read data and state-of-the-art SV analysis on individuals of diverse backgrounds, there are relatively few individuals (n=35) represented, and the overall callset is small compared to gnomAD or deCODE. As a result, there were very few SVs in our catalog that had an equivalent or similar SV in HGSVC that was not already found in gnomAD and/or deCODE and, therefore, the inclusion of HGSVC in this analysis has had negligible impact on the results presented in our manuscript. We refer the reviewer to the intersections of our callset with each individual annotation.

Actions: HGSVC is included when comparing our SV callset to existing annotations; results are presented in updated **Fig2e**.

3.3. Furthermore, there is an estimate that 2,884 SVs per individual are NCIG-only. This and the corresponding statistic is influenced by the cohorts this callset is compared against.

This analysis and the reported result have been updated after refining our SV intersection criteria (see **3.4**, below) and inclusion of the HGSVC annotation (see **3.2**, above).

Actions: Number of SVs reported as potentially exclusive to Indigenous Australians updated.

3.4. The 80% reciprocal similarity is too high for tandem repeats, or reflects an arbitrary value depending on how different alleles are counted. This may overstate the amount of novel variation in these genomes. One potential metric is to compare the number of all novel SV to only Alu MEI. The Alu insertions are well defined (typically),

and can help gauge the extent that novel variation is influenced by this high reciprocal overlap. Furthermore, the number of distinct SV (not near other SV) should be distinguished from SV at the same locus. One approach is to identify tandem repeat loci in CHM13 and apply a special merging of variants per locus, rather than using the reciprocal overlap.

We agree with the reviewer that our singular use of ‘80% reciprocal similarity’ to define whether an SV in our callset was ‘found’ / ‘not found’ in previous annotations is overly simplistic, especially for tandem repeat variants. To address this, we have refined the analysis to classify SVs based on their degree of overlap/similarity with previously annotated SVs. SVs in our callset are now classified as having high (>80%), moderate (50-80%), low (0-50%) or no (0%) reciprocal similarity to an annotated SV in gnomAD, deCODE and/or HGSVC (see 3.2, above). This updated analysis is presented in Fig2e.

As the reviewer predicted, different SV types behave somewhat differently during this analysis. There are higher proportions of SVs with ‘moderate’ and ‘low’ similarity to previously annotated SVs for tandem repeats and short-tandem repeats than for other SV types. These SVs were previously reported as ‘novel’ or ‘not found’. However, they may be better described as being new alleles of previously annotated variable nucleotide tandem repeat (VNTR) loci. We refer the reviewer to our new analysis comparing SVs of different types to existing annotations, which has also been included in Extended Data Fig4d.

Actions: Updated criteria used to assess similarity of SVs in our catalog to annotated SVs; results are presented in updated Fig2e and Extended Data Fig4d.

3.5. The number of unique SV should be counted as done in the current methods, but also only on the SV that are not proximal to others (within a few kb, or not near the same annotated tandem repeat on CHM13). The authors should clearly state what is a new SV: does this include different alleles on the same locus?

As suggested by the reviewer, we have refined the criteria used to compare SVs in our callset to existing SV annotations (see 3.4, above). The analysis now discriminates between completely novel SVs (i.e. no overlap to annotated SV) vs different alleles at the same locus (i.e. low or moderate reciprocal similarity).

We refer the reviewer to the following paragraph, where the analysis comparing our SV callset to existing annotations is described, following refinements/updates described above (**3.2 - 3.5**):

Of the 90,578/159,912 SVs that were successfully lifted to hg38, we found a highly similar annotated SV (>80% reciprocal overlap) for 37,421 and an annotated SV at the same position with moderate similarity (50-80%) for 22,625 (**Fig2e**). The latter were especially common for TR- and STR-associated SVs, where alternative alleles at variable TR/STR loci often appear as partially overlapping SVs (**Extended Data Fig4c,4d**). Taken together, this finds an annotated SV for just ~38% of all non-redundant SVs in our callset (**Fig2e**), with the remaining SVs that were successfully lifted having low (0-50%; $n=8,770$) or no (0%; $n=21,762$) overlap with any annotated variant. Because SVs that could not be lifted to hg38 cannot be assessed in this manner, we instead provide an upper-bound novelty estimate of 62% (assuming non-lifted SVs are all novel) and a lower-bound estimate of 19%.

Actions: Updated criteria used to assess similarity of SVs in our catalog to annotated SVs; results are presented in updated **Fig2e** and **Extended Data Fig4c,d**.

3.6. The read-based calling using cute-SV is subject to noise in calling SV near tandem repeats. An alternative to this is using de novo assembly to first assemble genomes, then call variants from assemblies. This is the approach taken by both the HGSVC and HPRC.

We appreciate this suggestion for a future avenue of analysis and we agree that an assembly-based approach might improve the accuracy of SV characterisation at tandem repeat loci. This is an approach we hope to implement in future Australian Indigenous genomics projects, guided by haplotype-resolved Indigenous reference genome assemblies, which we are developing in a separate study. However, for the present study we preferred an alignment-based approach for two main reasons: (i) alignment-based SV detection is less susceptible to variability in read length and read depth between samples, which is an important consideration in our study, given the variable quality of DNA obtained from saliva samples collected in remote Indigenous communities; and (ii) assembly-based methods are computationally-expensive by comparison to alignment-based methods, making it difficult for us to scale these to hundreds (and eventually thousands) of individuals.

Actions: No actions required.

3.7. Some analysis of the variant calling accuracy should be performed for the data used. Since ONT is continually evolving it is important to provide a benchmark for the study. The overall results for variant calls per individual are within expectation. Still, the results should begin with 2-3 sentences for validation of the sequencing and variant calling strategy on the HG002 reference. The diploid reference from <https://www.nature.com/articles/s41586-022-05325-5> can be used as ground truth, and the number of variants called from the reference should be compared to the alignment+cutesv pipeline at 15-50X coverage (by 5), in order to study the relationship between coverage and variant calls a the range of coverages generated on this cohort.

As suggested, we have added a new analysis in which we benchmarked our SV detection strategy using ONT data generated from the HG002 reference individual, which we originally sequenced for quality assessment purposes. Data were generated with both LSK110/R9.4.1 and LSK114/R10.4.1 ONT chemistry, and both were included in this analysis. We iteratively downsampled both datasets to represent a range of read-depth levels and evaluated SV detection by comparison to the HGSVC (Freeze 4) annotation for HG002 (taken here as a 'truth set'). We note that this analysis has been conducted recently in a benchmarking study by Harvey et al, which is currently in preprint¹¹, and we chose to use the same SV classification methodology and coverage range as this study. Similarly to Harvey et al, we observed: (i) improvement in SV recall up to ~20X coverage, beyond which recall plateaued, and; (ii) relatively little change in precision, regardless of coverage depth. We also observed no difference in SV detection performance between LSK110/R9.4.1 and LSK114/R10.4.1 ONT chemistry.

Since almost all (>96%, 136/141) of our samples were sequenced to >20X depth, this analysis confirms the suitability of our SV detection strategy and indicates that the coverage variability between samples in our study is not expected to have a major impact on SV detection. We refer the reviewer to the results of this benchmarking experiment, which has also been included in **Extended Data Fig2a**.

Actions: SV detection strategy benchmarked using HG002 sample at various coverage depths; results are presented in updated **Extended Data Fig2a**.

3.8. While long reads enable improved SV discovery over short reads, there are some analyses that are missing from this study that could improve its impact:

1. Copy number analysis. Cute-SV does not detect copy number variants (and is particularly bad at discovering rearrangements in general). It is straight forward to identify copy number variable genes through excess mapped read depth.

As suggested, we have applied a read-depth strategy (using *CNVpytor*) to characterise large copy-number variants (CNVs; including both duplications and deletions) across our cohort. This complements our existing analysis with *CuteSV*, which is not ideal for identifying large SVs (>50 kb). We have integrated CNV analysis into multiple sections throughout the revised manuscript, including: ‘Landscape of genomic structural variation’ section; ‘Distribution and diversity’ section; and ‘Functional context’ section. CNV analysis is presented in multiple new **Extended Data Figs (3a, 3d, 4a, 4b, 6e, 6f, 7, 9b, 9c)**. CNV analysis is described in the **Methods** section, and is mentioned in the **Abstract**.

Actions: New analysis of copy-number variation has been integrated throughout the manuscript and figures.

3.9. 2. Fixation index (F_{ST}) or V_{ST} can be calculated on this cohort relative to HPRC+HGSC data on non-tandem repeat loci when compared to the decode calls.

As suggested, we have used a Fixation Index (F_{ST}) method to assess genetic differences between Indigenous and non-Indigenous Australians and between different Indigenous communities in our study (**Extended Data Fig6a**). This analysis reiterates the clear genetic distinctions between Indigenous vs non-Indigenous groups, which were evident in the PCoA analysis presented in **Fig4a**. Relative pairwise distances between groups are broadly similar for F_{ST} (right panel below) and PCoA (left panel below) analyses:

In addition to this analysis on our own cohort, we attempted to directly compare communities in our study to the HGSVC cohort, using all SV calls on non-tandem-repeat loci via either F_{ST} and PCoA methods. However, this analysis is strongly confounded by various factors: (i) the use of different reference genomes (hg38 in HGSVC vs chm13 in our study), requiring liftOver of SVs prior to F_{ST} analysis; (ii) significant differences in sample collection, extraction and sequencing methodologies used by HGSVC vs our study; (iii) different bioinformatics approaches to SV detection used by HGSVC and our study; etc. We refer the reviewer to the outcome of this attempted F_{ST} /PCoA analysis, which shows excessive separation of HGSVC individuals from individuals in our study as a result of the combined ‘batch effect’ of the technical differences above. Given this major caveat, we have not included this comparison in our revised manuscript.

Actions: New analysis of genetic distances between communities using Fixation index (F_{ST}) was performed; F_{ST} results are reported in **Extended Data Fig6a**.

3.10. 3. A comparison of SNP and SV calls in this population. The fraction of population specific SNVs can be used to contrast the unique SVs, and can help give context for how accurate the estimate of population specific SVs.

Although not discussed in our study, all individuals were also analysed by short-read whole-genome sequencing, which is the subject of another manuscript currently under review at *Nature*. At the reviewer’s suggestion, we have used this data to call SNVs and assess the degree of population-specific genetic variation, in a manner that is completely independent from the SV analyses presented in our manuscript. The outcomes are broadly similar, with 22.36% of SNVs found in NCIG communities being absent from leading annotations (gnomAD, dbSNP, Simons Genome Diversity Project), compared to 19.09% (lower bound estimate) NCIG-exclusive SVs, and 46.37% of SNVs being found in only a single NCIG community, compared to 56.45% for SVs. We refer the reviewer to the results of this SNV analysis, which has also been included in **Extended Data Fig6c,d**.

c NCIG SNVs shared between communities**d NCIG SNVs comparison to existing annotation**
Actions: Population-specific genetic variation assessed using SNVs, as independent comparison to SVs; results presented in **Extended Data Fig6c,d**.

3.11. 4. Assemblies of HLA regions could give insight to innate/adaptive immunity in this population.

We agree that interesting insights into innate/adaptive immunity may arise from analysis of HLA regions among our cohort. However, we have a **strict ethical obligation to only undertake research that addresses questions on which the partner communities have been consulted and provided approval**. Unfortunately, HLA genotyping and implications for innate/adaptive immunity were not considered during community consultations, and fall outside the scope of approved research for this project (the focus of which was on characterizing genomic structural variation and STR expansion disorders).

Actions: Suggested analysis is out of scope of approved research.

REFERENCES

- Clarkson, C. *et al.* The archaeology, chronology and stratigraphy of Madjedbebe (Malakunanja II): A site in northern Australia with early occupation. *J. Hum. Evol.* **83**, 46–64 (2015).
- The process, biotic impact, and global implications of the human colonization of Sahul about 47,000 years ago. *J. Archaeol. Sci.* **56**, 73–84 (2015).
- Malaspinas, A.-S. *et al.* A genomic history of Aboriginal Australia. *Nature* **538**, 207–214 (2016).
- David Reich (Of Harvard Medical School). *Who We are and how We Got Here: Ancient DNA and the New Science of the Human Past*. (Oxford University Press, 2018).
- Hiscock, P. *Archaeology of Ancient Australia*. (Routledge, 2007).
- Arthur, W. S. & Morphy, F. *Macquarie Atlas of Indigenous Australia: Culture and Society Through Space and Time*. (Macquarie Library, Macquarie University, 2005).
- Beyter, D. *et al.* Long-read sequencing of 3,622 Icelanders provides insight into the role of structural variants in human diseases and other traits. *Nat. Genet.* **53**, 779–786 (2021).
- Otsuki, A. *et al.* Construction of a trio-based structural variation panel utilizing activated T lymphocytes and long-read sequencing technology. *Commun Biol* **5**, 991 (2022).
- Wu, Z. *et al.* Structural variants in the Chinese population and their impact on phenotypes, diseases and population adaptation. *Nat. Commun.* **12**, 6501 (2021).
- Kim, D., Song, L., Breitwieser, F. P. & Salzberg, S. L. Centrifuge: rapid and sensitive classification of metagenomic sequences. *Genome Res.* **26**, 1721–1729 (2016).
- Harvey, W. T. *et al.* Whole-genome long-read sequencing downsampling and its effect on variant calling precision and recall. *bioRxiv* 2023.05.04.539448 (2023) doi:10.1101/2023.05.04.539448.

Reviewer Reports on the First Revision:

Referees' comments:

Referee #4 (Remarks to the Author):

The analysis in response to the review was thorough and excellent; possibly the best I've ever seen.